**RESEARCH**

# Unraveling the phylogenomic diversity of Methanomassiliicoccales and implications for mitigating ruminant methane emissions

Fei Xie[1,2] , Shengwei Zhao[1,2], Xiaoxiu Zhan[1,2], Yang Zhou[1,2], Yin Li[1,2], Weiyun Zhu[1,2], Phillip B. Pope[3,4], Graeme T. Attwood[5], Wei Jin[1,2]* and Shengyong Mao[1,2]*

*Correspondence:
jinwei@njau.edu.cn;
maoshengyong@njau.edu.cn

[1] Ruminant Nutrition and Feed Engineering Technology Research Center, College of Animal Science and Technology, Nanjing Agricultural University, Nanjing, China
[2] Laboratory of Gastrointestinal Microbiology, Jiangsu Key Laboratory of Gastrointestinal Nutrition and Animal Health, National Center for International Research on Animal Gut Nutrition, College of Animal Science and Technology, Nanjing Agricultural University, Nanjing, China
[3] Department of Animal and Aquacultural Sciences, Faculty of Biosciences, Norwegian University of Life Sciences, Ås, Norway
[4] Faculty of Chemistry, Biotechnology and Food Science, Norwegian University of Life Sciences, Ås, Norway
[5] AgResearch Limited, Grasslands Research Centre, Palmerston North, New Zealand

## Abstract

**Background:** Methanomassiliicoccales are a recently identified order of methanogens that are diverse across global environments particularly the gastrointestinal tracts of animals; however, their metabolic capacities are defined via a limited number of cultured strains.

**Results:** Here, we profile and analyze 243 Methanomassiliicoccales genomes assembled from cultured representatives and uncultured metagenomes recovered from various biomes, including the gastrointestinal tracts of different animal species. Our analyses reveal the presence of numerous undefined genera and genetic variability in metabolic capabilities within Methanomassiliicoccales lineages, which is essential for adaptation to their ecological niches. In particular, gastrointestinal tract Methanomassiliicoccales demonstrate the presence of co-diversified members with their hosts over evolutionary timescales and likely originated in the natural environment. We highlight the presence of diverse clades of vitamin transporter BtuC proteins that distinguish Methanomassiliicoccales from other archaeal orders and likely provide a competitive advantage in efficiently handling $B_{12}$. Furthermore, genome-centric metatranscriptomic analysis of ruminants with varying methane yields reveal elevated expression of select Methanomassiliicoccales genera in low methane animals and suggest that $B_{12}$ exchanges could enable them to occupy ecological niches that possibly alter the direction of $H_2$ utilization.

**Conclusions:** We provide a comprehensive and updated account of divergent Methanomassiliicoccales lineages, drawing from numerous uncultured genomes obtained from various habitats. We also highlight their unique metabolic capabilities involving $B_{12}$, which could serve as promising targets for mitigating ruminant methane emissions by altering $H_2$ flow.

**Keywords:** Methanomassiliicoccales, Comparative genomics, Ruminants, Methane emissions, $B_{12}$

## Introduction

Methane ($CH_4$) has a strong global warming potential and the process by which it is produced (methanogenesis) is a key process underlying climate change [1]. Methanogenesis is largely constrained to a group of anaerobic methanogenic archaea that belong to the archaeal phylum Euryarchaeota, while several uncultured lineages in the candidate phyla Bathyarchaeota and Verstaraetearchaeota have also been found to possess methanogenesis-related gene clusters [2–4]. The phylum Euryarchaeota contains a large number and diversity of methanogen lineages [5, 6], which utilize a limited number of substrates via several methanogenesis pathways, including hydrogenotrophic, acetoclastic, methylotrophic, and the recently discovered alkylotrophic metabolism [7–11]. Methylotrophic methanogens use methylated compounds (e.g., methylamines, methanol, and methyl sulfides), occurring in two modes based on the presence or absence of cytochromes. Methylotrophs have received comparatively little attention owing to a prior perception that they were restricted to specific environments [12]. However, this has shifted in recent years following the widespread discovery of the seventh order of methanogens, known as Methanomassiliicoccales [13, 14].

The Methanomassiliicoccales order was previously described as "Rice Cluster C" or "Rumen Cluster C" and is distantly related to Thermoplasmatales [15–17]. They have been discovered in various anoxic environments, including wetlands, oceans, deep subsurface, rice crops, and the gastrointestinal tracts (GITs) of humans and several animals [12, 15, 18, 19]. Paul et al. originally characterized this group by observing an animal-associated cluster distinct from the environmental cluster through the analysis of comparative 16S rRNA and methyl-coenzyme M reductase alpha subunit (*mcrA*) genes [15]. Söllinger et al. have subsequently surveyed the physiology and ecology of Methanomassiliicoccales, highlighting their widespread distribution in global wetlands and animal GITs, as well as their significant contribution to methane emissions from these climate-relevant ecosystems [12]. The first Methanomassiliicoccales isolate, *Methanomassiliicoccus luminyensis*, was reported by Dridi et al. and is said to have been obtained from human feces [20]. In the past decade, despite many efforts, only eight complete genomes of Methanomassiliicoccales isolated from the GITs of various animals have been characterized through pure cultures, including those from humans [21, 22], termites [23], chickens, sheep [24, 25], and cattle. The cultivated representatives of Methanomassiliicoccales have demonstrated a distinct energy metabolism from that of other methylotrophic methanogens [5, 23, 26] and have been found to utilize methylamines and methanol as electron acceptors [13, 15, 20, 21, 27, 28]. Genomic analyses conducted thus far have revealed that Methanomassiliicoccales lack cytochromes and encode a truncated Wood-Ljungdahl (WL) pathway [11, 23, 26, 29]. In this context, methyl groups cannot be oxidized to $CO_2$; instead, they obligately employ $H_2$ to reduce methyl compounds to $CH_4$ [10, 11]. Similar to *Methanimicrococcus blatticola* and *Methanosphaera* species [5], Methanomassiliicoccales rely on external $H_2$ and have become a model for $H_2$-dependent methylotrophic methanogens due to their extensive distribution [12].

The genomes of these cultured methanogens have yielded fundamental insights into the biology of Methanomassiliicoccales [26, 29–32]. Nevertheless, research on members of the order Methanomassiliicoccales has progressed slowly owing to difficulties in their isolation and cultivation process, and constraints such as employment of strict anaerobic conditions

needed for their study [33]. The phylogenetic taxonomy, functional characterization, and metabolic roles of Methanomassiliicoccales remain inadequately characterized and have been restricted to a few isolated strains and genomes [26, 30, 34]. This is despite recent transformations in metagenomic technologies that have facilitated the generation of hundreds of thousands of metagenome-assembled genomes (MAGs) from diverse habitats, including numerous archaeal methanogens [35–41]. Several studies employing genomic analyses of selected MAGs have revealed that many previously uncultured lineages are indeed $H_2$-dependent methylotrophic methanogens and have initiated efforts to assess the diversity of taxa that are challenging to cultivate [3, 42, 43].

The occurrence of Methanomassiliicoccales in the human gut has been linked to lowered trimethylamine oxide (TMAO) production and cardiovascular disease risk [34, 44]. While in other animals, such as ruminants, in which the metabolism of methanogens contributes around 37% of the world's total anthropogenic methane emissions [27, 45], the Methanomassiliicoccales order has been relatively poorly characterized [17]. A global rumen survey conducted by Henderson et al. previously suggested that ruminal methanogens is limited to the phylum Euryarchaeota [46]. However, over the last decade, Thermoplasmatales-affiliated sequences (i.e., Methanomassiliicoccales) have been found in a variety of ruminant species and often constitute the second most abundant group of methanogens found in the rumen microbiome [17, 27, 47–52]. More recently, Shi et al. conducted a comparative analysis of the ruminal archaeal microbiota in sheep with low and high methane emissions and showed that *Methanosphaera* spp., which shares an energy metabolism similar to Methanomassiliicoccales, exhibited elevated abundance in sheep with low methane emissions [53]. Methanomassiliicoccales was not included in that study due to a lack of genomic information at the time of analysis [53]. Overall, the role of Methanomassiliicoccales in the rumen remains unclear due to a lack of sufficient phylogenetic classification, and further analysis specifically focused on Methanomassiliicoccales is needed in light of this previous research deficiency.

Here, we collected 208 publicly available Methanomassiliicoccales genomes and MAGs from various global habitats as well as reconstructed 33 high-quality Methanomassiliiccales-affiliated MAGs. MAGs were recovered from 370 metagenomic datasets that were generated across 10 GIT regions of seven different ruminant species and eight metagenomic samples enriched with trimethylamine from cow rumen fluids. On our newly acquired genome atlas, we applied a large-scale genome-resolved comparison to reveal the essential metabolic functions of Methanomassiliicoccales and clarify the evolutionary classification of Methanomassiliicoccales originating from the GITs of ruminants. Subsequently, a genome-centric isolation strategy and in vitro enrichments of samples collected from the goat rumen and cow abomasum allowed us to obtain pure culture strains of the major Methanomassiliicoccales groups. Finally, we additionally integrated metagenomic and metatranscriptomic data from Shi et al. [53] to provide novel insights into the distinct Methanomassiliicoccales groups in the rumen and their roles in methanogenesis.

## Results

### The phylogenetic position and evolutionary signatures of Methanomassiliicoccales

To determine the phylogenetic position of the relatively new order Methanomassiliicoccales, we investigated the evolutionary divergence of the Euryarchaeota using

publicly available datasets from the National Center for Biotechnology Information (NCBI) database, including 146 representative genomes from 13 orders of Euryarchaeota and 46 outgroup genomes from Crenarchaeota (Additional file 2: Table S1). Methanonatronarchaeales were excluded from this study because of a lack of records of their complete genomes. A phylogenetic tree constructed from 26 concatenated ribosomal proteins showed that these archaeal orders belonged to distinct lineages (Fig. 1A). Among these orders, Methanomassiliicoccales shared a common ancestor with non-methanogenic Thermoplasmatales, appeared later than Methanosarcinales, and had an evolutionary origin independent of the class I and II methanogens (Fig. 1A). Methylotrophs with cytochromes can oxidize additional methyl groups to $CO_2$, and this capability is exclusively found among members of

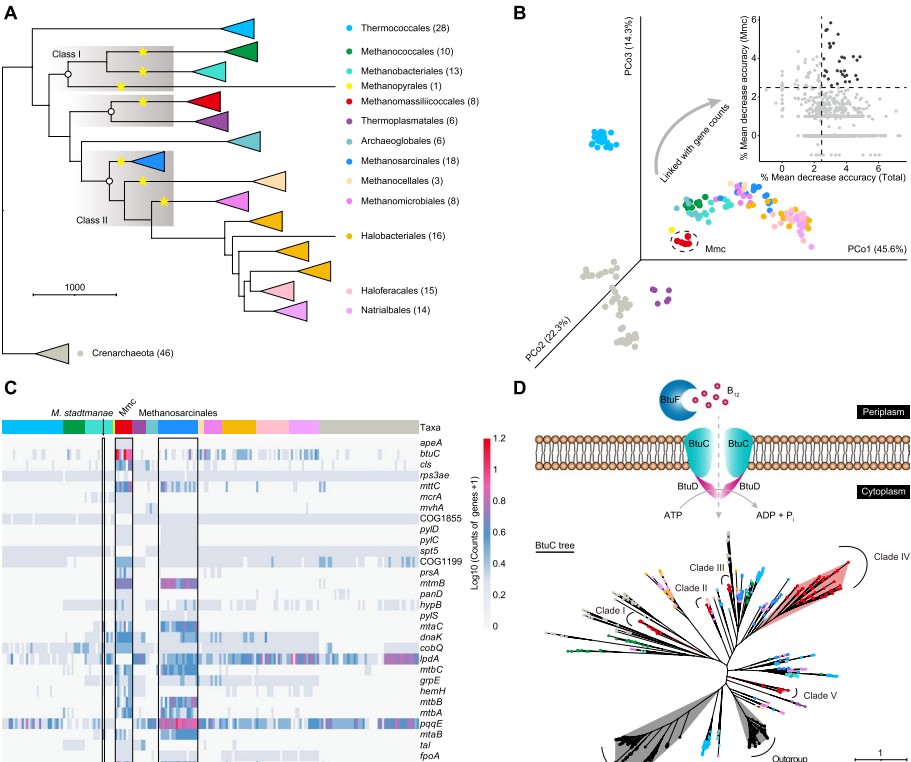

**Fig. 1** Evolutionary genomic analyses of Methanomassiliicoccales. **A** A phylogeny of 146 complete Euryarchaeota genomes and 46 outgroup taxa from the phylum Crenarchaeota (represented by different colored circles) was reconstructed based on 26 ribosomal proteins. The gray background represents the topological divergence of the three monophyletic clades, and the yellow stars represent the known methanogenic Euryarchaeota (except for Methanonatronarchaeales). The numbers in the brackets represent the number of genomes corresponding to each order. **B** PCoA depicting the genomic divergence based on a predicted function presence/absence matrix determined from 192 genomes colored by their taxonomy. The random forest classifier defined the importance score for each putative gene in Methanomassiliicoccales to the overall variance. **C** Heatmap constructed from a gene count matrix of 31 important genes identified by random forest analysis. The upper colored strips represent the genome classification in each column, and the right side shows the ranked genes. **D** Schematic showing a functional $B_{12}$ acquisition system, including the periplasmic binding protein BtuF and ABC transporter BtuCD. The phylogenetic tree of BtuC was constructed based on the alignments of 762 protein sequences with 281 aligned positions. The branches are colored according to the taxonomic source of these sequences, and the assumed five clades, along with the outgroup, are labeled on the phylogenetic tree. Mmc, Methanomassiliicoccales; *M. stadtmanae*, *Methanosphaera stadtmanae*

the Methanosarcinales [10]. While in methylotrophs without cytochromes, methyl groups cannot be oxidized to $CO_2$; instead, they obligately use $H_2$ to reduce methyl compounds to $CH_4$ [11]. In this context, we observed a homolog of HdrE, which encodes the cytochrome *b*-containing membrane anchor of the heterodisulfide reductase complex (HdrDE) responsible for receiving electrons from methanophenazine, was present in all Methanosarcinales genomes and in the species *Methanosphaerula palustris* within Methanomicrobiales (Additional file 1: Fig. S1). However, Methanomassiliicoccales, Methanocellales (Rice Cluster I; despite the prior discovery of cytochromes in some of the MAGs [54]), and other methanogenic orders, all lacked the HdrE subunit (Additional file 1: Fig. S1A).

Next, we examined the individualized genotypic traits of Methanomassiliicoccales and applied homology searches to generate protein annotations, which were then summarized in a gene presence or absence matrix. A gene-based principal coordinate analysis (PCoA) was conducted to visualize the relationships among the different members of Euryarchaeota. In the PCoA plot, the genomes of the 13 Euryarchaeota orders appeared to cluster according to their taxonomic assignments, with Methanomassiliicoccales exhibiting prominent separation (Fig. 1B). To identify differential indicator genes, the overall variances were defined by a random forest classifier (Fig. 1C and Additional file 3: Table S2). Among the top 31 significantly differential genes, eight genes encoding methyltransferases (*mttB*, *mttC*, *mtbA*, *mtbB*, *mtbC*, *mtmB*, *mtaB*, and *mtaC*) that enable the activation of methylated compounds were enriched in *Methanosphaera stadtmanae*, Methanomassiliicoccales, and Methanosarcinales (Fig. 1C). Notably, *M. stadtmanae* belongs to the Methanobacteriales order, but its energy metabolism is similar to that of Methanomassiliicoccales, as it relies on a $H_2$-dependent methylotrophic metabolism [5]. Indeed, we observed that *M. stadtmanae* possesses the *mtaB* and *mtaC* genes, which facilitate methanogenesis from methanol and $H_2$ (Fig. 1C). Several biological processes related to the amino acid pyrrolysine biosynthesis and cobalamin ($B_{12}$) transport were identified to potentially facilitate the adaptation of Methanomassiliicoccales. Previous studies have indicated that pyrrolysine may be an evolutionary indication of methylotrophic archaea [9], with the monomethylamine methyltransferase (MtmB) from *Methanosarcina barkeri* found to contain pyrrolysine [55]. We found that Methanomassiliicoccales and Methanosarcinales uniquely possessed the genes *pylCD* and *pylS*, which are responsible for synthesizing pyrrolysine from two lysines (Fig. 1C). Moreover, *btuC*, which encodes the vitamin $B_{12}$ transporter BtuC, was abundant in Methanomassiliicoccales but rare in other orders (Fig. 1C and Additional file 1: Fig. S1B). Previous research has confirmed that multiple transporters in *Bacteroides thetaiotaomicron* exhibit distinct preferences for corrinoids and contribute to microbial adaptability in the human gut [56]. To explore the diversity of BtuC in Methanomassiliicoccales, a protein tree was constructed based on BtuC homology from various public sources. The results revealed the presence of five distinct BtuC clades within Methanomassiliicoccales, which were spread among other orders (Fig. 1D). Methanomassiliicoccales may therefore have diverse competitive advantages in the presence of structurally diverse $B_{12}$. Another cofactor-related gene, *pqqE*, which encodes PqqA peptide cyclase that catalyzes the synthesis of vitamin-like pyrroloquinoline quinone, was

absent in Methanomassiliicoccales (Fig. 1C). Likewise, the gene-encoding dihydroli-poyl dehydrogenase (*lpdA*), a mitochondrial enzyme involved in energy metabolism, was absent in Methanomassiliicoccales but common in other orders (Fig. 1C).

### Phylogenomic diversity and reclassification of Methanomassiliicoccales

Advances in metagenomics have enabled genome-centric identification of many previously uncultured species that represent the potential tree of life lineages [35]. To further classify the Methanomassiliicoccales within the order, we collected 208 corresponding genomes from globally available datasets and additionally reconstructed 33 high-quality MAGs as well as two cultured complete genomes that were collectively recovered from ruminant GITs (Additional file 4: Table S3). These 243 genomes ranged from 1 to 2.66 Mb, with an average completeness of 90% and contamination of 1.2% (Fig. 2A and Additional file 4: Table S3). Based on a concatenated set of 118 archaeal marker proteins, we constructed a genome-wide phylogenetic tree that showed the broad categorization of Methanomassiliicoccales into environmental and gastrointestinal clades (Fig. 2A), which is consistent with previous 16S rRNA and *mcrA* gene sequencing data [30, 32, 34]. The strains isolated from anthropogenic samples (e.g., reactor mud, fuel cells, and industrial wastes) are mainly sourced from the natural environment or the GITs of animals. Therefore, we observed that anthropogenic genomes have been incorporated into both environmental and gastrointestinal clades (Fig. 2A). We also found that the origin of the taxa did not always align with the clades to which they had been artificially assigned. For instance, two MAGs retrieved from groundwater and deep subsurface microbial communities were incorporated in the gastrointestinal clades (Fig. 2A and Additional file 4: Table S3). In contrast, a branch whose genomes were retrieved from human feces, including the cultured Issoire-Mx1 strain, was clustered with the environmental clades (Fig. 2A and Additional file 4: Table S3). Strong cohesiveness was also observed between specific branches and their hosts, such as humans, pigs, baboons, and cattle (Fig. 2A).

We extracted 16S rRNA genes from genomes to calculate sequence identity and pairwise estimated genome-wide average amino acid identity (AAI) and average nucleotide identity (ANI) for determining known or novel taxonomic units in Methanomassiliicoccales (Additional file 4: Table S3). Maximum likelihood trees were constructed using 16S rRNA genes from 141 genomes, *mcrA* genes from 199 genomes, and whole genomes (*n* = 243), which collectively support the Methanomassiliicoccales order comprising of one family with significant variation at the genus level (Fig. 2B and Additional file 1: Fig. S2). In this study, we reclassified 22 distinct genera and 105 species based on the AAI and ANI at the 65 and 95% recognition level (Fig. 2C and Additional file 1: Fig. S3). Genomes assigned to the recognized genera, *Methanomassiliicoccus* (G. 9 and G. 10) and *Methanoplasma* (G. 19 and G. 20), were reassigned to two genera, whereas G. 21 represented all *Methanomethylophilus* members (Fig. 2C). Based on the phylogenetic tree, 12 (G. 1 to G. 12) and 10 genera (G. 13 to G. 22) were assigned to the environmental and gastrointestinal clades, respectively (Fig. 2C). Among these genera, 17 had more than two genomes, and five comprised a single genome, representing low-abundance or environment-specific taxa (Fig. 2C). The novel genus G. 22 consisted of the largest number of genomes (67 genomes), which were frequently recovered from gastrointestinal metagenomic assemblies (Fig. 2C). In G. 22, 96% of the genomes were MAGs reconstructed

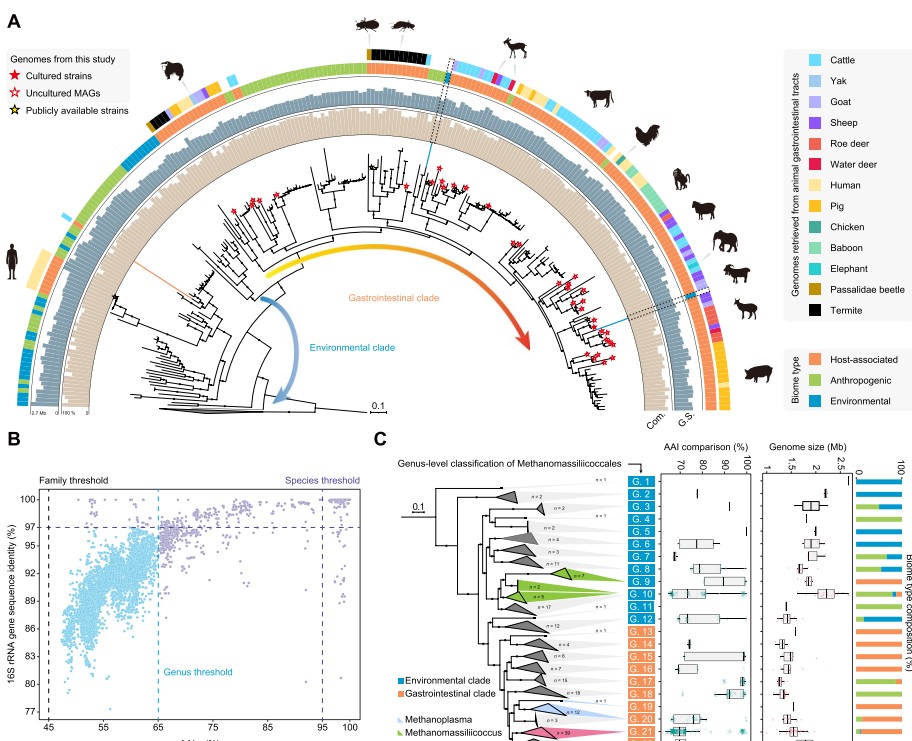

**Fig. 2** Methanomassiliicoccales taxon defined based on whole-genome identity pairwise comparisons. **A** A maximum likelihood tree based on 243 Methanomassiliicoccales genomes, including 33 MAGs and two cultured strains from this study, was inferred from a concatenated set of 118 archaeal marker proteins. The gray triangle corresponds to the five outgroup taxa from the order Thermoplasmatales. Support values > 80% are shown as black dots. Stars on the clade represent the genomes reconstructed in this study and publicly available cultured genomes, whereas unmarked clades represent MAGs retrieved from previous studies. Colored strips on the two outermost rims show the biome type to which the isolation belongs and the animal source of the host-associated genomes. Two layers of the concentric bar plot indicate the genome size (G.S.) and the corresponding assembled completeness (Com.). **B** Amino acid identity (AAI) values and 16S rRNA gene sequence pairwise comparisons showing acknowledged genus (AAI > 65%) and species (AAI > 95% or 16S rRNA gene sequence identity > 97%) cutoff values. **C** Acronyms represent the 22 Methanomassiliicoccales genera (G. 1 to G. 22) identified in this study. Clades in the phylogenomic tree were collapsed according to the taxonomic level. Colored clades represent three previously named genera. Genera were divided into the environmental and gastrointestinal clades. Box plots showing divergences in the AAI index and genome size between genomes assigned to these genera. The composition of the biome category in each genus is displayed in a histogram

from previously uncultured phylotypes, with only two cultured strains isolated from the chicken cecum (DOK strain) and sheep rumen (ISO4-G1 strain) (Additional file 4: Table S3). G. 10 had an average genome size of 2.21 Mb and a GC content of up to 61%, significantly larger than those of the other genera in Methanomassiliicoccales (Fig. 2C and Additional file 4: Table S3).

### Functional comparison and genomic expansion of Methanomassiliicoccales

To further investigate the functional properties, we conducted a pan-genome analysis of the taxonomic genera uncovered in the order Methanomassiliicoccales. Orthologs were discovered among the 243 genomes, and 12,093 orthologous genes were generated (Additional file 5: Table S4). A PCoA based on these orthologous gene profiles

showed that the genomes constituting the different genera appeared to cluster simultaneously depending on their taxonomic assignments and habitats (ANOSIM; R = 0.558, *P* = 0.001; Fig. 3A). Surprisingly, the three genomes from G. 1 and G. 2 were found to be closely related to five outgroup taxa (Thermoplasmatales), despite their taxonomic classifications in the Genome Taxonomy Database (GTDB; Fig. 3A) suggesting that they were members of Methanomassiliicoccales. Mapping gene orthologs and predicting gene gain and loss events onto a phylogenetic tree was based on 243 genomes (Fig. 3B). We found that Methanomassiliicoccales shared 1656 orthologous genes with their common ancestor, and another 5085 genes were defined as gained genes. Gene gain events broadly occurred in the different Methanomassiliicoccales genera and were especially frequent in G. 10, G. 21, and G. 22 (Fig. 3B). Additionally,

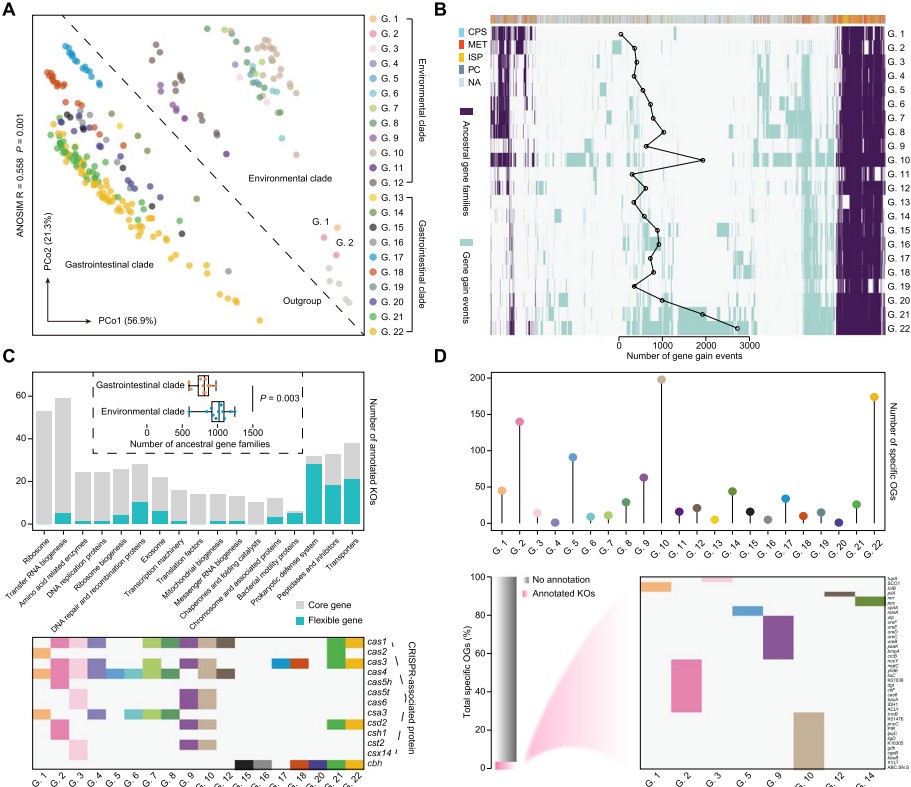

**Fig. 3** Pan-genome analysis of Methanomassiliicoccales. **A** PCoA showing the genomic divergence based on the gene orthologs of these 243 genomes plus five outgroup taxa from the order Thermoplasmatales and colored by their assigned genus. ANOSIM tests were used to assess the significance of dissimilarities between the genera calculated by the Bray-Curtis method. **B** A two-color heatmap showing the ancestral gene and gained events for each genus. The top lines represent the functional category of orthologous genes in each column and are colored according to the COG annotations. The number of gene gain events among different genera is shown on the line plot. CPS, cellular processes and signaling; MET, metabolism; ISP, information storage and processing; PC, poorly characterized; NA, no annotation. **C** The number of ancestral gene families in genera from the environmental and gastrointestinal clades were compared using Wilcoxon rank-sum test. Histograms showing KO annotations of core (≥ 80% of genera have) and flexible (< 50% of genera have) genes in the 1656 ancestral gene families of Methanomassiliicoccales. Genes encoding Cas proteins and choloylglycine hydrolase (*cbh*) are present in different genera. **D** The number of unique genes specific to each genus among the 968 gained genes is shown at the top, and their KO annotations are shown at the bottom. The heatmap of the 40 annotated genes distributed across 8 genera is expanded to the right

genera belonging to the gastrointestinal clades lost more ancestral genes than the environmental clades (Wilcoxon rank-sum test, $P = 0.003$; Fig. 3C).

Similar to other microbial taxa, most of the core ancestral genes encoded proteins involved in housekeeping biological processes, such as ribosomes, amino acid-related enzymes, and DNA replication (Fig. 3C). The prokaryotic defense system, peptidases and inhibitors, and transporters are part of the flexible ancestral gene pool. CRISPR–Cas modules are prokaryotic adaptive defense systems against foreign genetic elements [57]. We found that Methanomassiliicoccales genera from diverse communities varied in their probability of being invaded by viruses, and more Cas proteins were present in most of the genera from the environmental clades than in the gastrointestinal host-associated clades (Fig. 3C and Additional file 6: Table S5). Moreover, most gastrointestinal clades lacked 21 gene-encoding transporters, including five that encode proteins required to transfer small compounds across biological membranes (Additional file 1: Fig. S4). Nevertheless, an acclimatization marker, such as choloylglycine hydrolase or bile salt hydrolase, was specific to six genera belonging to the gastrointestinal clades (G. 15, G. 16, G. 18, G. 20, G. 21, and G. 22; Fig. 3C), indicating their early adaptation to the animal gut environment.

Further analyses revealed that 968 gained genes were specific to 22 genera, with 94% identified as unannotated proteins that could not be classified functionally (Fig. 3D). Notably, many unannotated proteins appeared only in G. 2, G. 10, and G. 22 (Fig. 3D and Additional file 7: Table S6), implying that they may have numerous unknown functions. The genes encoding multiple ureases (*ureBCDEF*) that hydrolyze urea into ammonia and $CO_2$ and a membrane-bound urea transporter (*utp*) were unique to G. 9 (Fig. 3D). Moreover, genes encoding $Cu^{2+}$ and $Mg^{2+}$ transport proteins (*nosY* and *mgtC*) were found in two MAGs derived from the tailing pond of G. 2 (Fig. 3D). Sugar-sensing (*trmB*) and degradation (xylosyltransferase and fructose-bisphosphate aldolase) genes were unique to G. 10 (Fig. 3D), which was likely related to their larger genome size, although most of the other specific genes (93.9%) in this genus had unknown functions.

### Reconstruction of the central metabolic pathways shared by Methanomassiliicoccales

The central pathways of the 243 genomes were predicted and reconstructed to assess the metabolic features of Methanomassiliicoccales (Fig. 4 and Additional file 8: Table S7). Previous studies have revealed that the WL tetrahydromethanopterin ($H_4$MPT) pathway is widespread in archaeal genomes and is most likely ancestral to archaea [58]. However, this pathway was not fully present in Methanomassiliicoccales because of the absence of the enzymes formylmethanofuran: $H_4$MPT formyltransferase (*ftr*) and methenyl-$H_4$MPT cyclohydrolase (*mch*), which catalyze the second and third steps of the $H_4$MPT methyl branch, respectively (Fig. 4). Additionally, most of the environmental clade contained tetrahydromethanopterin S-methyltransferase (*mtrA* and *mtrH*), which catalyzes the formation of methyl-coenzyme M (Additional file 1: Fig. S5). Interestingly, another methyl branch of the WL pathway that converts formate to acetyl-CoA using tetrahydrofolate ($H_4$F) was found in the genomes of Methanomassiliicoccales (Fig. 4). Furthermore, the gastrointestinal clades reduced methylamines, methanol, and methylated sulfides (methanethiol and dimethylsulfide) for methanogenesis, whereas the environmental clades were largely methanol-specific (Additional file 1: Fig. S5). Core

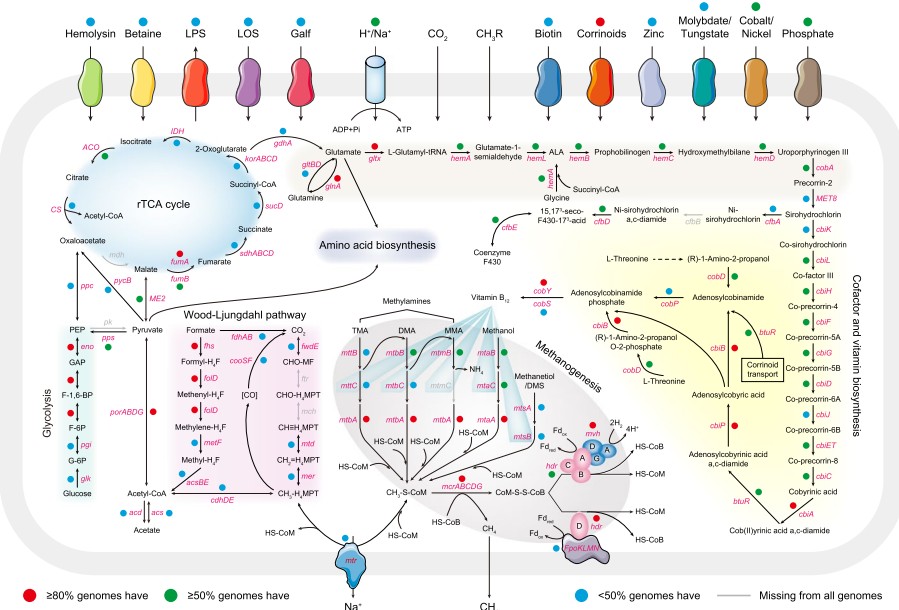

**Fig. 4** Overview of metabolic potentials in Methanomassiliicoccales. The detected pathways and genes in the 243 Methanomassiliicoccales genomes related to glycolysis, the rTCA cycle, amino acid biosynthesis, methanogenesis, cofactor, and vitamin biosynthesis, Wood-Ljungdahl pathway, and various transporters. Different background colors represent different metabolic modules. The rays emitted by vitamin $B_{12}$ represent the genes encoding cobamide-dependent functions. Colored circles and gray lines indicate the frequency of the genes present in the 243 genomes. Detailed gene information in each pathway is presented in Additional file 8: Table S7. GAP, glyceraldehyde-3-phosphate; PEP, phosphoenolpyruvate; MF, methanofuran; $H_4$MPT, tetrahydromethanopterin; CoA, coenzyme A; CoB, coenzyme B; CoM, coenzyme M; DMS, dimethylsulfide; Fd, ferredoxin; ALA, δ-aminolevulinate

methanogenesis genes (*mcrABCDG*, *mtbA*, and *mtaA*) were also found in Methanomassiliicoccales, whereas the *mtmC* gene, which encodes an enzyme that catalyzes the transfer of a methyl group from monomethylamine to monomethylamine-specific corrinoid proteins, was missing from all genomes (Fig. 4). We found that all the Methanomassiliicoccales genomes lack the homologs of the HdrE subunit (*hdrE*) but carry the Fpo-like subunits (*fpoKLMN*), which may directly interact with the HdrD subunit (*hdrD*) to form the energy-converting ferredoxin:heterodisulfide oxidoreductase complex (Fig. 4). Moreover, Methanomassiliicoccales possessed a large amount of methyl viologen-dependent hydrogenase (*mvhADG*) and the Hdr complex (*hdrABC*) to couple the reduction of the heterodisulfide CoM-S-S-CoB and a ferredoxin with $H_2$ via flavin-based electron bifurcation (Fig. 4). $B_{12}$ is an important coenzyme for microorganisms and their surrounding communities, and different organisms are selective towards particular $B_{12}$ vitamins due to their structural diversity [59]. We found that the genes involved in the anaerobic $B_{12}$ biosynthesis pathway synthesizing adenosylcobalamin from precorrin-2 and the cobalt transport system (*cbiOMQN*) were largely encoded by Methanomassiliicoccales, except for several genes (*MET8*, *cbiK*, *cbiJ*, *cobP*, and *cobS*) that were missing in half of the genomes (Fig. 4). Notably, the complete BtuFCD components responsible for $B_{12}$ transport were also widespread in different Methanomassiliicoccales genera (Additional file 1: Fig. S5). $B_{12}$ is a major component of corrinoid proteins, forming the central core of methyltransferases (*mtaC*, *mttC*, *mtbC*, *mtmC*, and *mtsB*) [59]. We also

observed that these $B_{12}$-dependent methyltransferases were more prevalent in the gastrointestinal clades (G. 18, G. 20, G. 21, and G. 22), with the exception of the human gut-derived environmental clade G. 9.

**Assembly and cultivation of Methanomassiliicoccales from ruminant GITs**

Despite being a source of methane emissions, Methanomassiliicoccales in ruminant GITs are poorly characterized [60]. We, therefore, reconstructed 33 high-quality MAGs belonging to Methanomassiliicoccales specifically using the metagenomic data from 370 samples across 10 GIT regions of seven different ruminant species and eight samples enriched with trimethylamine (TMA) from cow rumen fluids (Fig. 5A and Additional file 4: Table S3). Most of the MAGs were from the rumen, with six assembled from the abomasum, the true acid stomach in ruminants. The MAG genome sizes ranged from 1 to 2.18 Mb (average 1.66 Mb), and they encoded an average of 1585 genes with an average gene length of 824 bp (Additional file 4: Table S3). The genomes were well-curated, and 15 MAGs were evaluated as near-complete with genome completeness ranging from 92 to 99% and almost no contamination (average 1.8%) with other genome fragments (an average of 3 rRNAs and 44 tRNAs were detectable) (Additional file 4: Table S3). A phylogenetic tree based on a total of 68 Methanomassiliicoccales genomes from ruminant GITs showed that these 33 MAGs formed six distinct genera, including G. 15, G. 16, G. 17, G. 20, G. 21, and G. 22, and were assigned to 23 species (Fig. 5B and Additional file 4: Table S3). The most abundant genera were G. 21 and G. 22, indicating their significance in methane production in ruminants.

Based on the results of the GIT metagenomic assemblies, the sources of the rumen and abomasum microbiota were supplemented with TMA to enrich for Methanomassiliicoccales members that remained viable in laboratory cultures (Fig. 5A). Two strains were successfully isolated and cultured from goat rumen (named LGM-RCC1) and cow abomasum samples (named LGM-DZ1); their phylogenetic positions belonged to the genera G. 22 and G. 21, respectively (Fig. 5B). Both strains possessed irregular coccoid cells and the ability to thrive on methanol and methylamines but methanogenesis was obligately dependent on $H_2$ (Fig. 5C). Rumen-derived LGM-RCC1 was able to grow with methyl-3-methylthiopropionate (Fig. 5C), which was also reported in the strain ISO4-H5 isolated from the sheep rumen [24]. Different substrate preferences were also evident between the two strains, with LGM-RCC1 and LGM-DZ1 showing better growth with TMA and methanol, respectively (Fig. 5C). LGM-DZ1 exhibited greater methanogenesis efficiency, producing the maximum amount of methane in 1 week, whereas LGM-RCC1 required 1 month (Fig. 5C). pH testing showed that both strains could grow to a maximum pH of 8, whereas LGM-DZ1 could tolerate a lower pH of 5.5. The optimal temperature for LGM-RCC1 was between 35 and 40 °C, while that for LGM-DZ1 was between 40 and 45 °C.

The complete genomes of LGM-RCC1 (1,664,211 bp) and LGM-DZ1 (1,768,094 bp) contained 1691 and 1739 predicted protein-coding genes, 44 and 43 tRNAs, and four rRNA clusters, respectively (Additional file 4: Table S3). Based on the COG and KEGG annotations, LGM-DZ1 had more functional categories than LGM-RCC1 in certain genomic features (Fig. 5D and Additional file 9: Table S8). For example, proteins involved in the transport and metabolism of amino acids, inorganic ions, coenzymes,

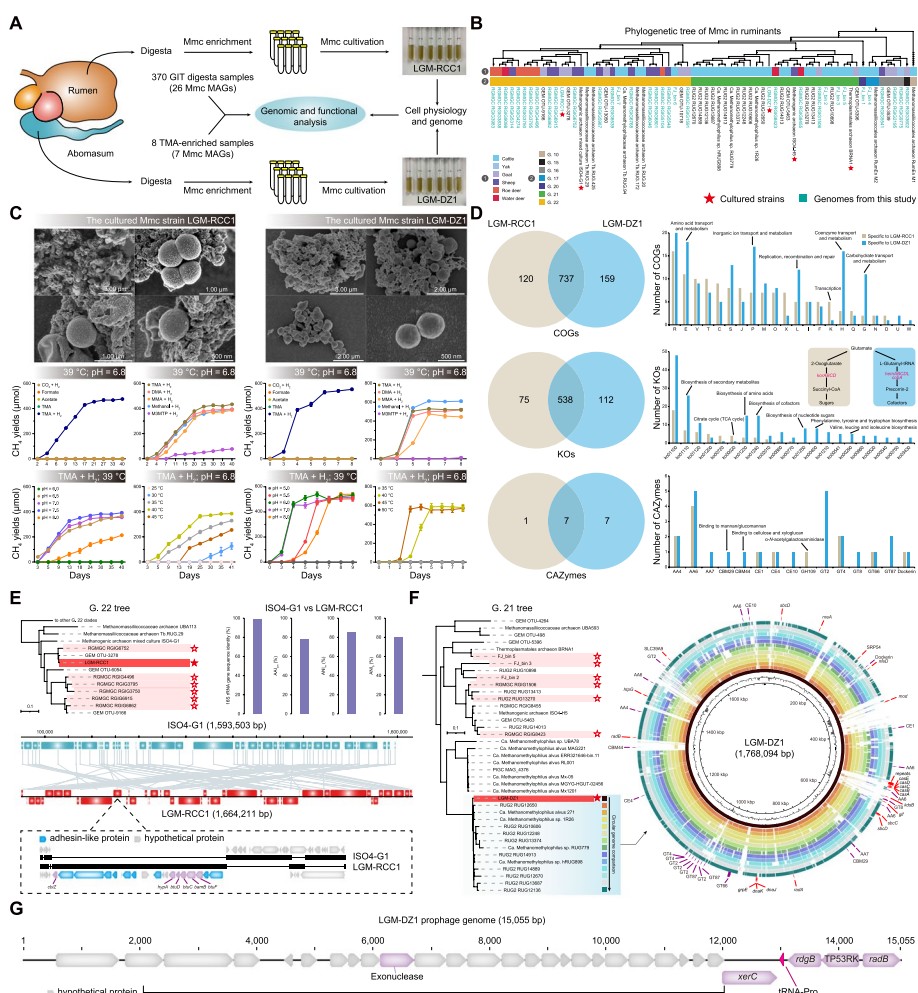

**Fig. 5** Methanomassiliicoccales genomes reconstructed from the ruminant GITs. **A** Schematic depicting an overview of the workflow for this section. **B** A phylogenetic tree was constructed using 68 Methanomassiliicoccales genomes derived from ruminant GITs. **C** Genome features of Methanomassiliicoccales strains LGM-RCC1 and LGM-DZ1. Cell imaging of the two strains was performed using an electron microscope. Line plot showing methane production by two strains under different nutrients, substrates, pH, and temperature. **D** Comparison of the levels of functional modules (COGs, KOs, and CAZymes) of the two strains. The right bar charts show the categories of the functional modules in the specific sets, and the major enriched categories are labeled. **E** Phylogenetic trees of G. 22 genera extracted from the whole Methanomassiliicoccales tree are shown. Colored strips and stars represent the genomes reconstructed in this study, and the strain LGM-RCC1 is highlighted. The 16S rRNA gene sequence identity, AAI values, and ANI values ($ANI_m$ and $ANI_b$) between the strains ISO4-G1 and LGM-RCC1 and their genome-wide alignments are shown. **F** Circle visualization showing a pan-genome comparison of the strain LGM-DZ1 and 13 other genomes belonging to one clade. LGM-DZ1 was used as the reference, and the two inner rings show the DNA size and GC content of the reference genome. Colored outer rings show regions of the comparison genomes that matched the LGM-DZ1 genome. Genome names from inside to outside are listed at the bottom right. The location of the selected genes in the reference genome is denoted at the outer ring. **G** The genome structure of a prophage found in the strain LGM-DZ1 is shown at the bottom

and carbohydrates, as well as genomic replication, recombination, and repair, were highly enriched in LGM-DZ1 (Fig. 5D). The genes (*korABCD*) encoding the key enzymes in the reductive TCA (rTCA) cycle that converts succinyl-CoA to 2-oxoglutarate were enriched in LGM-RCC1, whereas various genes (*hemABCDL* and *cobA*)

involved in the synthesis of precorrin-2 from glutamate were enriched in LGM-DZ1 (Fig. 5D). In addition, an analysis using the carbohydrate-active enzyme (CAZyme) database showed that glycoside hydrolase GH109 (α-*N*-acetylgalactosaminidase), which degrades mucus layer sugars, was specific to LGM-RCC1 (Fig. 5D).

The phylogenomic position of LGM-RCC1 was closely related to another cultured strain, ISO4-G1, which was previously isolated from the sheep rumen [25] (Fig. 5E). The strains had a 98.4% 16S rRNA gene sequence similarity; however, genome-wide comparisons using AAI (77.9%) and ANI (85.7% of ANIm) did not indicate they belonged to the same species (Fig. 5E). Whole-genome alignment revealed that the genome of LGM-RCC1 has undergone large-scale evolutionary divergence, with genomic rearrangements and inversions (Fig. 5E). A structural variant occurred in a locally co-linear block of the LGM-RCC1 genome with the ISO4-G1 genome, where LGM-RCC1 aggregated multiple adhesin-like proteins (ALPs; Fig. 5E). The complete $B_{12}$ transport system encoded by *btuC*, *btuD*, and *btuF* was adjacent to these ALPs (Fig. 5E). Moreover, LGM-DZ1 formed an independent branch with 13 MAGs (most of these genomes are from the rumen) and well-represented the genomic content of these uncultured genomes based on genomic alignment (Fig. 5F). Genes involved in the resistance to acid stress, including the signal recognition particle subunit SRP54 and those involved in the molecular chaperone-repair system (*dnaK*, *dnaJ*, and *grpE*), were identified in LGM-DZ1 and the clustered MAGs (Fig. 5F and Additional file 9: Table S8). Hence, these genes may protect intracellular metabolism [61]. Furthermore, CRISPR and adjacent Cas proteins (*casABCDE*) in the LGM-DZ1 genome (Fig. 5F) could provide sequence-specific recognition and protection in the presence of phages. However, a 15,055-bp prophage sequence was also found in the LGM-DZ1 genome, which represented the first virus to infect members of Methanomassiliicoccales and was identified as a novel species containing a large number of unknown proteins in the current database (Fig. 5G).

### Community role of Methanomassiliicoccales in sheep methane emissions

We retrieved public metagenome and metatranscriptome sequencing data previously generated from the rumens of low-methane-emitting (LME) and high-methane-emitting (HME) sheep to understand the methane emissions from ruminants [53]. The total gene expression of bacterial and archaeal communities in the LME and HME sheep showed opposite trends, with high expression in archaea when methane production was high and in bacteria when methane production was low (Additional file 1: Fig. S6A). Further analysis of the methanogenic lineages at the order level showed a significant increase in gene expression of Methanobacteriales, Methanomassiliicoccales, Methanomicrobiales, Methanococcales, and Methanonatronarchaeales in the HME sheep (Fig. 6A and Additional file 1: Fig. S6B). Focusing specifically on the Methanobacteria, Shi et al. revealed higher relative abundances of organisms belonging to *Methanobrevibacter* spp. in the HME sheep [53]. However, due to a lack of Methanomassiliicoccales genome representation at the time of this study, their transcriptional activity in the LME and HME animals was not recorded. Therefore, we further aligned transcriptomic reads from the LME and HME sheep to 68 ruminant-derived Methanomassiliicoccales genomes and seven publicly available genomes from the *Methanobrevibacter* genus. The expression profiles of each genome were calculated based on the percentage of mapped RNA reads (Additional

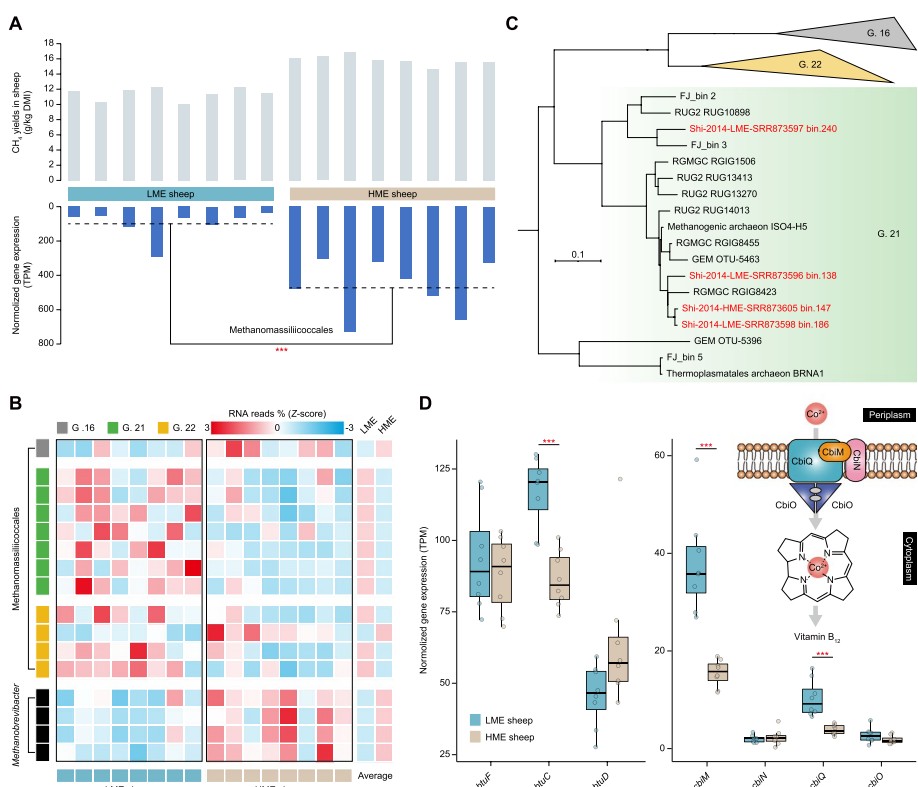

**Fig. 6** Metagenomic and metatranscriptomic analysis of the rumen microbiome from the LME and HME sheep. **A** The bar plot shows methane ($CH_4$) yields and the total gene expression of rumen Methanomassiliicoccales communities in the LME and HME sheep. Differences in gene expression between the two groups are compared. **B** A heatmap displaying the genomic expression of 10 Methanomassiliicoccales and four *Methanobrevibacter* genomes from G. 16, G. 21, and G. 22 significantly differing in the LME and HME sheep. The expression of these genomes between the two groups was standardized using the *Z*-score method, and an average value was calculated. **C** A phylogenetic tree was constructed using Methanomassiliicoccales MAGs assembled from the LME and HME sheep and other Methanomassiliicoccales genomes. These four MAGs are presented in red font, and their respective sample sources are labeled. **D** Boxplots display the total gene expression of seven genes (*btuF*, *btuC*, *btuD*, *cbiM*, *cbiN*, *cbiQ*, and *cbiO*) related to the microbial B$_{12}$ and cobalt transport systems in the LME and HME sheep. Schematics of the microbial cobalt transport system are also included. The significance level between the two groups has been compared. Wilcoxon rank-sum test, \*\*\**P* < 0.001. DMI, dry matter intake; TPM, transcripts per million

file 10: Table S9). Indeed, we reaffirmed that four genomes from *Methanobrevibacter* were differentially enriched by more than 2-fold in the HME sheep (Fig. 6B). In contrast, 12 significantly differential Methanomassiliicoccales genomes (*P* < 0.05) from G. 16, G. 21, and G. 22 were mostly enriched in the LME sheep, except for RGIG2195 (G. 16) and ISO4-G1 (G. 22) with 1.5- and 2.8-fold enrichment in the HME sheep, respectively (Fig. 6B). Shi et al. previously reported a higher relative abundance of metabolically similar *Methanosphaera* spp. in the LME sheep [53]. However, at a transcriptomic level, we observed numerically higher gene expression levels in *M. stadtmanae* in the LME sheep, although the differences were not statistically significant (Additional file 1: Fig. S6B). To advance this analysis further, we reanalyzed metagenomic data from both the LME and HME groups recovering 17 high-quality archaeal MAGs affiliated to *Methanosphaera* (*n* = 1), Methanomassiliicoccales (*n* = 4, classified as *Methanomethylophilus* in the GTDB),

and *Methanobrevibacter* ($n = 12$) (Additional file 4: Table S3). The phylogenomic positions of these four Methanomassiliicoccales MAGs maintained taxonomic consistency all clustering within G. 21 (Fig. 6C). In support of our hypothesis that G. 21 Methanomassiliicoccales populations are more active in the LME sheep, three of the four MAGs were obtained from the LME samples, signifying adominance (Fig. 6C).

We also identified significant correlations between Methanomassiliicoccales and other microorganisms at the order level (|Spearman's correlation coefficient| > 0.8 and $P <$ 0.001). This included 22 positive correlations, involving 15 bacterial orders, five fungal orders, one Ciliophora order, and one archaeal order, as well as three negatively correlated orders (Additional file 1: Fig. S6C). The relationship between the rumen microbiome and methane emissions is complex as microorganisms frequently interact at the molecular level by competing or sharing nutrients [62]. In this context, we subsequently found clues that the genes *cbiM* and *cbiQ*, which encode the transmembrane permease proteins of the cobalt transport system, were highly expressed in the LME sheep (Wilcoxon rank-sum test, $P < 0.001$; Fig. 6D). Cobalt plays a vital role in the biosynthesis of the corrin ring of vitamin $B_{12}$ [59]. In light of this, we examined the $B_{12}$ biosynthesis pathway in the rumen microbiome and observed elevated levels of gene expression in multiple processes (*gltB*, *gltD*, *glnA*, *MET8*, *hemA*, and *cbiL*) within the LME sheep (Additional file 1: Fig. S6D). Furthermore, we also observed a significant increase in the total gene expression of the vital $B_{12}$ transporter BtuC (*btuC*, $P < 0.001$; Fig. 6D) in the rumen microbiome of the LME sheep.

## Discussion

Methanomassiliicoccales are well-known for their wide distribution across global environments and distinctive $H_2$-dependent methyl-reducing methanogenesis process. They belong to a recently defined order within the Euryarchaeota phylum, with limited genomic data available. As a result, their classification primarily depends on individual genes, and functional capabilities are limited to a small number of cultured strains. In this study, we collected genomes and MAGs from various habitats, reassembled high-quality MAGs from the GITs of diverse ruminant animals, cultivated new methanogenic strains of Methanomassiliicoccales, and conducted an integrated comparative genomic analysis to gain novel insights into this order.

The genome-wide phylogenetic tree constructed using completed genomes from the Euryarchaeota phylum revealed that the methanogenesis capabilities of these methanogens have evolved gradually throughout their evolutionary history [63, 64]. Methanomassiliicoccales, which evolved independently and lack cytochromes, emerged at a relatively later point in time compared to Methanosarcinales and do not fall into either the class I or class II methanogens. As previously discussed, methanogens possessing cytochromes are evolutionarily younger than those lacking cytochromes [5]. These findings may suggest that the acquisition of cytochromes in methanogens was possibly driven by environmental adaptation. This is supported by the presence of the cytochrome *b* subunit HdrE in the species *Methanosphaerula palustris*, which is the only methanogen, besides Methanosarcinales, known to encode it [65].

Comparisons between Methanomassiliicoccales and other archaeal orders revealed a high level of differentiation in genes related to methanogenesis and cofactor metabolism.

Gene variation for methyltransferases within the genomes of *M. stadtmanae*, Methanomassiliicoccales, and Methanosarcinales may be indicative of convergent evolution between these different clades since they share parallel traits adapted to contrasting environments. The observation of the abundant $B_{12}$ transporter BtuC proteins in Methanomassiliicoccales genomes may imply that other archaeal orders may have a lower reliance on $B_{12}$. Previous research has reported that $B_{12}$ selectivity in microorganisms is crucial in the context of microbial interactions [59]. Hence, it can be hypothesized that distinct BtuC clades may confer competitive advantages to Methanomassiliicoccales in the presence of structurally diverse $B_{12}$ molecules.

A genome-wide phylogenetic tree, which included the genomes of cultured organisms and uncultured MAGs from various habitats, suggested that Methanomassiliicoccales are more diverse in their community distribution than previously considered. Söllinger et al. previously found that Methanomassiliicoccales were subdivided in two broad phylogenetic clades designated environmental and GIT clades based on PCR targeting of their 16S rRNA and *mcrA* genes [12, 23, 30]. Based on an extensive phylogenomic tree, we have found that some genomes do not always align with their assigned clades. Additionally, we discovered that the gastrointestinal clade has demonstrated the potential for fidelity with its hosts over evolutionary timescales. We reclassified the Methanomassiliicoccales order and suggested that it exhibits an underestimated level of taxonomic differentiation. Among these newly identified genera, G. 22 is a genus primarily composed of MAGs obtained from the GITs of animals, indicating their specialization towards host GITs. Notably, a larger genome and higher GC content in G. 10 may suggest that members of this genus acquired more genes to adapt to their environments.

The ancestral and gained genes were further predicted to explore the evolutionary divergence among different Methanomassiliicoccales genera. We found that G. 10, G. 21, and G. 22 carried the most abundant gene gain events, allowing them to adapt and survive their surrounding conditions. By comparing the richness of ancestral genes, we can elucidate the inheritance or loss of ancestral information among different genera, thereby reflecting their evolution and origins [66]. Our findings showed that genera within the gastrointestinal clades have lost more ancestral genes compared to those in the environmental clades, indicating that the Methanomassiliicoccales associated with hosts originated from the natural environment. Moreover, Methanomassiliicoccales that survive in natural environments are likely more resistant to viral invasion, which may be associated with microbial density and diversity [57].

The findings from functional analysis suggested that the evolutionary divergence between genera may signify the emergence of novel genes. The presence of numerous unannotated proteins in G. 2, G. 10, and G. 22 underscores the need for further investigations to uncover the roles and significance of these proteins in the context of their respective genera. We also found that G. 9 clustered with the environmental clade, while all of its members were derived from the human gut. The unique presence of genes related to urease metabolism in G. 9 suggested that they may have selectively acquired the ability to assimilate urea to adapt to intestinal conditions. Moreover, the identification of genes associated with $Cu^{2+}$ and $Mg^{2+}$ transport proteins in MAGs from G. 2 is likely a result of horizontally transferred events from surrounding bacteria under prolonged harsh environmental stress, aimed at maintaining their cellular stability [65, 66].

Overall, genetic variability has led to functional partitioning and ecological divergence among Methanomassiliicoccales lineages, resulting in distinctive metabolic capabilities crucial for their respective ecological niches.

The central metabolic pathways shared by cultured genomes and uncultured MAGs of Methanomassiliicoccales revealed the presence of an incomplete WL $H_4MPT$ pathway [58]. Interestingly, the genes (*mtrA* and *mtrH*) responsible for catalyzing the formation of methyl-coenzyme M were found within the environmental clades. Given the scarcity of environmentally cultured Methanomassiliicoccales strains, we speculate that some of the environmental strains may also depend on methanogenesis from $H_2$ and $CO_2$. We also observed a distinction between gastrointestinal and environmental clades in terms of substrate preferences, with the former predicted to reduce methylamines, methanol, and methylated sulfides for methanogenesis, while the environmental clades were methanol-specific. This was likely due to the richer nutrient composition in the GITs compared to other environments. The absence of the *mtmC* gene (monomethylamine corrinoid protein) from all genomes is noteworthy and we speculate that there are likely alternative proteins that play a similar role since the ability of Methanomassiliicoccales to grow using monomethylamine has been demonstrated experimentally [23].

Methanomassiliicoccales were found to possess genes involved in the anaerobic $B_{12}$ biosynthesis pathway, which is significant for their metabolic processes. Furthermore, the widespread presence of complete BtuFCD components for $B_{12}$ transport indicated a capability to acquire and utilize this essential coenzyme. Like other nutrient metabolites studied in microbial interactions, this also implies that the transport of $B_{12}$ from the external environment is crucial for Methanomassiliicoccales members. The possession of de novo biosynthesis and salvage routes for $B_{12}$ in Methanomassiliicoccales is not surprising, as methylotrophic corrinoid proteins (*mtaC*, *mttC*, *mtbC*, *mtmC*, and *mtsB*) are homologous to the core corrinoid-binding domain of $B_{12}$ proteins [59, 67]. This suggests that it is an important precursor for $B_{12}$-dependent methyltransferases production by Methanomassiliicoccales during methanogenesis. Interestingly, the prevalence of $B_{12}$-dependent methyltransferases varied among different Methanomassiliicoccales genera, possibly reflecting their specific metabolic requirements related to $B_{12}$ in their ecological niches.

There is a growing interest in discovering targets to mitigate methane emissions in the GITs of ruminants, thereby reducing energy loss from the animals and lessening the environmental footprint of ruminant farming [68, 69]. We reconstructed 33 high-quality Methanomassiliicoccales MAGs using metagenomic data from natural gastrointestinal samples collected from various ruminant species, as well as TMA-enriched samples. An analysis of these genomes and their prevalence within the GITs of ruminants revealed the presence of six distinct genera, with G. 21 and G. 22 emerging as significant contributors to methane production in ruminants. Furthermore, we successfully isolated and cultured two new strains of Methanomassiliicoccales, one belonging to G. 21 and the other belonging to G. 22, providing insights into the metabolic diversity within the Methanomassiliicoccales genera. The genomic comparison between LGM-RCC1 and LGM-DZ1 highlighted differences in their functional categories, many of which were explained by environmental selection. The phylogenetic analysis also revealed the close relationship between

LGM-RCC1 and another cultured strain, ISO4-G1. Despite their high 16S rRNA gene sequence similarity, genome-wide comparisons indicated they did not belong to the same species, presenting substantial evolutionary divergence. For example, the complete $B_{12}$ transport system, encoded by *btuC*, *btuD*, and *btuF*, was adjacent to the ALPs in LGM-RCC1 genome. Such gene clusters in LGM-RCC1 may indicate a specialized capture mechanism for exogenous $B_{12}$. Additionally, LGM-DZ1 was the first strain isolated from the abomasum and established that Methanomassiliicoccales could survive in the acidic environment of the digestive tract. It formed an independent branch with 13 MAGs, although interestingly the majority of them were derived from the rumen. LGM-DZ1 and related uncultured MAGs exhibited genes associated with acid stress resistance, as well as CRISPR–Cas systems for protection against phages. These findings highlight that LGM-DZ1 has a unique survival mechanism in the abomasum of ruminants.

Methanogens play a significant role in the rumen ecosystem by efficiently consuming $H_2$, thereby promoting the recycling of reduced cofactors and facilitating the breakdown and fermentation of plant materials [70, 71]. Therefore, strategies for mitigating methane emissions should consider alternative pathways for $H_2$ utilization to decrease methanogenesis levels while preserving rumen function [69, 72–74]. Our re-analysis of the dataset previously described by Shi et al. [53] to incorporate Methanomassiliicoccales gene expression in sheep categorized as either LME or HME showed that the overall Methanomassiliicoccales gene expression was higher in the HME sheep; however, for several Methanomassiliicoccales genera, such as G. 21 (or *Methanomethylophilus*) and G. 22, their gene expression was in fact higher in the LME sheep. Interestingly, our re-analysis also observed higher abundance of *M. stadtmanae*, which are also $H_2$-dependent methylotrophic methanogens, in the LME sheep [53]. The genera with similar metabolic characteristics are likely to serve as indicative microbes for an LME microbiome and may be applied in future breeding programs for ruminants.

Our aforementioned genomic comparisons conducted in Methanomassiliicoccales revealed that $B_{12}$ biosynthesis and transport pathways are prevalent across various genera. Since $B_{12}$ serves as a crucial coenzyme mainly synthesized by bacteria [59], we speculate that this higher genomic capability to metabolize $B_{12}$ may confer a competitive advantage to the Methanomassiliicoccales in the LME sheep. The increased expression of cobalt transporters in the LME sheep supports the idea that a substantial number of components required for $B_{12}$ biosynthesis can be efficiently transported into microbial cells in the rumen when methane emissions are low. In correspondence, we also observed that multiple genes involved in the $B_{12}$ biosynthesis pathway in the rumen microbiome were upregulated in the LME sheep. Furthermore, the elevated gene expression of the BtuC transport protein in the LME sheep suggests that Methanomassiliicoccales-affiliated organisms with a higher demand for $B_{12}$ are more active under low-methane production conditions. $B_{12}$ interactions among rumen microbes could serve as a crucial mechanism, enabling specific members of Methanomassiliicoccales to inhabit ecological niches and thereby alter the direction of $H_2$. This points towards the prospect of nutrient availability (such as $B_{12}$) playing an important role in regulating methane emissions in ruminants [59, 75–77].

## Conclusions

Our study indicates numerous undefined lineages within Methanomassiliicoccales primarily detected through current uncultured MAGs, highlighting their substantial phylogenetic diversity. Genetic variability among Methanomassiliicoccales genera has led to functional partitioning and ecological divergence, resulting in distinctive metabolic capabilities crucial for their respective ecological niches. The gastrointestinal clades of Methanomassiliicoccales have demonstrated the potential for fidelity with their hosts over evolutionary timescales and likely originated in the natural environment. The central metabolic pathways shared by cultured genomes and uncultured MAGs of Methanomassiliicoccales revealed that $B_{12}$ is a crucial cofactor for Methanomassiliicoccales in the utilization of methyl compounds during methanogenesis. Additionally, an integrated genome-centric metatranscriptomic analysis revealed that several genera of Methanomassiliicoccales were more transcriptionally active in low-emitting animals, where elevated gene expressions of $B_{12}$ synthesis and transport may be associated with greater $B_{12}$ sharing in the LME sheep. Overall, these findings provide novel insights into the evolutionary history, metabolic cycling, and community function of the significant methanogenic order Methanomassiliicoccales.

## Materials and methods

### Sample acquisition and metagenomic sequencing

We observed in preliminary experiments that trimethylamine (TMA) served as a more effective substrate for the enrichment of Methanomassiliicoccales. Therefore, Methanomassiliicoccales were enriched with TMA from cow rumen fluids using a modified BRN medium as described in the subsequent culture section [78]. Eight mixed enrichments from different periods were gathered for metagenomic sequencing. We also collected 370 natural digesta samples from the gastrointestinal tracts (GITs) of ruminant animals, including dairy cattle, water buffalo, yak, goat, sheep, roe deer, and water deer, for metagenomic sequencing to obtain additional ruminant-associated Methanomassiliicoccales genomes. Sample collection and microbial DNA extraction were performed as described in our previous study [40]. In summary, all fresh samples were collected and stored at −80 ℃, and subsequently, DNA was extracted from each sample (~200 mg) following the protocol from Yu and Morrison [79] based on a repeated bead-beating. Metagenomic libraries of 350 bp insert sizes were constructed using the TruSeq DNA PCR-Free Library Preparation Kit (Illumina, San Diego, CA, USA) following the manufacturer's instructions and sequenced on an Illumina NovaSeq 6000 platform with paired-end 150 bp.

### Metagenome assembly, binning, and genome collection

All data from the eight enriched and 370 GIT samples were processed using the same metagenomic analysis pipeline. Adapters and low-quality reads from the raw data were trimmed using Trimmomatic [80] (v.0.33), and then potential DNA contamination (including that from the hosts, plants, and human) was removed by mapping the sequence data to the closest NCBI genome using BWA-MEM [81] (v.0.7.17). Specifically, we used the genome sets of hosts, including dairy cattle (*Bos Taurus*,

GCA_002263795.2), water buffalo (*Bubalus bubalis*, GCA_003121395.1), yak (*Bos mutus*, GCA_000298355.1), goat (*Capra hircus*, GCA_001704415.1), sheep (*Ovis aries*, GCA_002742125.1), roe deer (*Capreolus pygargus*, GCA_000751575.1) and water deer (*Hydropotes inermis*, GCA_006459105.1); plants, including sorghum (*Sorghum bicolor*, GCA_000003195.3), wheat (*Triticum aestivum*, GCA_002220415.3), robur (*Quercus robur*, GCA_900291515.1), sweet potato (*Ipomoea batatas*, GCA_002525835.2), medicago (*Medicago truncatula*, GCA_000219495.2), rice (*Oryza sativa*, GCF_000005425.2), barley (*Hordeum vulgare*, GCA_900075435.2), maize (*Zea mays*, GCA_003185045.1 and GCA_000005005.6), soybean (*Glycine max*, GCA_000004515.4), and ryegrass (*Lolium perenne*, GCA_001735685.1); and human (*Homo sapiens*, GCA_000001405.28), as references for mapping to decrease the potential DNA contamination. The remaining reads from each sample were assembled individually using MEGAHIT [82] (v.1.1.1; parameters: -min-contig-len 500 -presets meta-large) and IDBA-UD [83] (v.1.1.3; parameter: -pre_correction -min_contig 500 -mink 90 -maxk 124), then merged using Minimus2 [84] (AMOS, v.3.1.0). Reads from the same GIT regional samples in each ruminant species were co-assembled using MEGAHIT [82] (v.1.1.1). The assembled contigs were corrected for single bases, insertions, and deletions based on remapped reads using BWA-ALN [81] (v.0.7.17) and SAMtools [85] (v.1.9). Short contigs (<1.5 kb) were removed. Each metagenomic assembly was binned according to the base distribution and coverage depth using MaxBin [86] (v.2.2.4), MetaBAT2 [87] (v.2.11.1), and CONCOCT [88] (v.1.1.0) with default parameters. MAGs constructed using different software were integrated and refined using the DAS tool [89] (v.1.1.1) and dereplicated using dRep [90] (v.2.5.4; parameter: -pa 0.95 -sa 0.99 -cm larger) with a 99% ANI cutoff. The completeness and contamination of the non-redundant MAGs were assessed using CheckM [91] (v.1.2.1; parameter: lineage_wf), and the genome sizes were corrected and estimated based on their completeness and contamination [92]. Taxonomic annotations were obtained using alignments with the Genome Taxonomy Database (release 207; http://gtdb.ecogenomic.org/) in the GTDB-Tk [93] (v.2.1.0) toolkit. Based on a previous study on Methanomassiliicoccales [34], the MAGs belonging to the Methanomassiliicoccales order with substantial completeness and low contamination levels ($\geq$70% completeness and <5% contamination) were included in this study. Other genomes from humans, non-ruminant animals, and the global environment, including those from the Earth's Microbiomes project [38], were retrieved from the NCBI publicly available database (https://www.ncbi.nlm.nih.gov/) or links provided by the authors. Genomic information for all the Methanomassiliicoccales genomes used in this study are summarized in Additional file 4: Table S3.

### Culture, growth conditions, and electron microscopy

The gastrointestinal digesta samples from the seven ruminant animals, including dairy cattle, water buffalo, yak, goat, sheep, roe deer, and water deer, were used for enriching and isolating Methanomassiliicoccales strains. Five milliliters of rumen fluids or 0.3 g of abomasum contents were diluted with 50 mL anaerobic diluent, and 0.5 mL of the mixture was transferred into a 10-mL BRN medium containing antibiotics, vitamins, and TMA solution, filled with $H_2$. Cells were serially cultured on an incubator shaker. Two

strains were successfully isolated and cultured from goat rumen and cow abomasum. The following is a detailed description:

### Material preparation

The modified BRN medium for methanogens contained (per 1000 mL) the following: tryptone, 2 g; yeast extract, 2 g; clarified rumen fluid, 100 mL; mineral solution no. 1, 50 mL; mineral solution no. 2, 50 mL; Balch trace elements (+ NST), 10 ml; fatty acids solution, 50 mL; resazurin solution (0.1%), 1 mL; $NH_4Cl$, 1 g; coenzyme M, 0.04 g; $NaHCO_3$, 5 g; and cysteine-HCl, 0.5 g; adjusted to pH 6.9–7.0. After inoculation, tubes were pressurized with $H_2$ to 100 kPa [78].

(1)  Mineral solution no. 1 (1000 mL) contained $K_2HPO_4$, 3.0 g.
(2)  Mineral solution no. 2 (1000 mL) contained $KH_2PO_4$, 3.0 g; $(NH_4)_2SO_4$, 6.0 g; NaCl, 6.0 g, $MgSO_4·7H_2O$, 0.6 g; and $CaCl_2·2H_2O$, 0.6 g.
(3)  Modified Balch trace elements (+ NST) (1000 mL) contained nitrilotriacetic acid, 1.5 g; $MgSO_4·7H_2O$, 3.0 g; $MnSO_4·H_2O$, 0.45 g; NaCl, 1.0 g; $FeSO_4·7H_2O$, 0.1 g; $CoSO_4·7H_2O$, 0.18 g; $CaCl_2·2H_2O$, 0.1 g; $ZnSO_4·7H_2O$, 0.18 g; $CuSO_4·5H_2O$, 0.01 g; $AlK(SO_4)_2·12H_2O$, 0.018 g; $H_3BO_3$, 0.01 g; $NaMoO_4·2H_2O$, 0.01 g; $NiSO_4·6H_2O$, 0.1 g; $Na_2SeO_4$, 0.19 g; $Na_2WO_2·2H_2O$, 0.1 g [94].
(4)  Fatty acid solution (1000 mL) contained acetic acid, 6.85 mL; propionic acid; 3.0 mL; butyric acid, 1.84 mL; 2-methylbutyric acid, 0.55 mL; isobutyric acid, 0.47 mL; valeric acid, 0.55 mL; and isovaleric acid, 0.55 mL.
(5)  Vitamins solution (1000 mL) contained 1,4-naphthoquinone, 0.25 g; D-Ca-pantothenate, 0.2 g; nicotinamide, 0.2 g; p-aminobenzoni acid, 0.025 g; riboflavin, 0.2 g; thiamine-HCl, 0.2 g; pyridoxine HCl, 0.2 g; biotin, 0.025 g; folic acid, 0.25 g; and cyanocobalamin, 0.025 g [95]. Vitamin solution was added before the use of the medium (0.1 mL per 10 mL medium).
(6)  The anaerobic diluting solution (1000 mL) contained mineral solution no. 1, 50 mL; mineral solution no. 2, 50 mL; $Na_2CO_3$, 3 g; cysteine-HCl, 1 g; and resazurin solution (0.1%), 1 mL.
(7)  The mixed antibiotics contained penicillin, 160,000 U/mL; streptomycin 200,000 U/mL; lincomycin, 100 mg/mL; vancomycin, 50 mg/mL; and colistin sulfate, 100 mg/mL. Each antibiotic was used 0.1 mL per 10 mL medium.
(8)  The methyl compounds contained methanol, 6.0 mol/L; monomethylamine hydorchloride, 6.0 mol/L; dimethylamine hydorchloride, 3.0 mol/L; trimethylamine hydrochloride, 2.0 mol/L; and methyl 3-methylthiopropionate, 6.0 mol/L. Each methyl compound was used 0.1 mL per 10 mL medium.

### Enrichment and isolation

Approximate 5-mL rumen fluids or 0.3 g abomasum contents (from goat and dairy cow, respectively) was 10 times diluted with anaerobic diluting solution. Then, 0.5 mL diluted sample was disposed into Hungate roll tubes (Bellco glass 28 mL, USA) contained 9.5 mL BRN medium. Penicillin, streptomycin, trimethylamine hydrochloride, and vitamin solutions had been added before inoculation. Then, tubes were pressurized with $H_2$ to

100 kPa. Subsequently, all tubes were incubated at 39 °C, 50 rpm/min shaking. Consecutive transfer was performed every 7 days to enrich the target methanogens. During enrichment, antibiotics of penicillin, streptomycin, lincomycin, vancomycin, and colistin sulfate were used alternatively to eliminate bacteria. Lumazine (2,4-Dihydroxypteridine, final concentration 0.025%) was used to depress the growth of *Methanobrevibacter* spp. in the enrichment culture [96, 97].

Continuous 10 times gradient dilution of the enrichment culture was performed to obtain a pure target strain. This procedure was carried out in a anaerobic workstation (Whitley A35 HEPA, UK). The purification assessment of strains followed a two-step process. First, DNA was extracted from the enrichment culture as mentioned above. Universal primers were used for PCR amplification of the 16S rRNA genes of both bacteria and methanogens. For bacterial amplification, the 8F (5′-CACGGATCCAGA GTTTGAT(C/T)(A/C) TGGCTCAG-3′) and 1510R (5′-GTGAAGCTTACGG(C/F) TACCTTGTTACGACTT-3′) primers were employed [98]. Methanogen amplification was carried out using the 86F (5′-GCTCAGTAACACGTGG-3′) and 1340R (5′-CGGTGTGTGCAAGGAG-3′) primers [99]. Subsequently, agarose gel electrophoresis was employed to confirm the absence of bacterial contamination in the amplification products. Second, the PCR products were purified using a PCR product purification kit (Vazyme, China), followed by insertion into the pESI-T vector using a cloning kit (Tsingke, China). These constructs were then transformed into competent cells (*Escherichia coli* DH5α) and cultured in LB medium. The transformed cells were spread onto LB agar supplemented with ampicillin and incubated at 37 °C for 10 h. Single colonies were selected for further cultivation in LB medium, and the resulting culture liquid was subjected to sequencing. The obtained sequencing results were compared and analyzed using Geneious [100] (v.2022.1.1) to ensure base consistency, with all sequences consistently identified as pure strains.

### Substrate utilization

Hungate roll tubes containning 9.5 mL BRN medium were used. Group 1, fill 20 kPa $CO_2$ and 80 kPa $H_2$; group 2, add formic acid (final concentration, 30 mmol/L); group 3, add acetic acid (final concentration, 30 mmol/L); group 4, add trimethylamine hydorchloride (final concentration, 20 mmol/L); group 5, add trimethylamine hydorchloride (final concentration, 20 mmol/L) and fill 100 kPa $H_2$. Pure strain cultures in exponential phase were used as inoculum, and 0.3 mL was inoculated into each tube. All tubes were incubated at 39 °C, 50 rpm/min shaking. Gas pressure, $CH_4$, and $H_2$ were measured in different time intervals.

### Effect of different methyl compounds on the methane production of strains

Hungate roll tubes containing 9.5 mL BRN medium were used. Group 1, add methanol (final concentration 60 mmol/L); group 2, add monomethylamine hydrochloride (final concentration 60 mmol/L); group 3, add dimethylamine hydrochloride (final concentration 30 mmol/L); group 4, add trimethylamine hydrochloride (final concentration 20 mmol/L); group 5, add methyl 3-methylthiopropionate (final concentration 60 mmol/L). Pure strain cultures in exponential phase were used as inoculum, and 0.3 mL

was inoculated into each tube. All tubes were filled 100 kPa $H_2$ and were incubated at 39 °C, 50 rpm/min shaking. Gas pressure, $CH_4$, and $H_2$ were measured in different time intervals.

### Effect of pH on the methane production of strains

Different pH BRN medium were prepared. Trimethylamine hydrochloride (final concentration 20 mmol/L) was added as substrate. All tubes were filled 100 kPa $H_2$ and incubated at 39 °C, 50 rpm/min shaking. Gas pressure, $CH_4$, and $H_2$ were measured in different time intervals.

### Effect of temperature on the methane production of strains

Trimethylamine hydrochloride (final concentration 20 mmol/L) and 100 kPa $H_2$ were added as substrates. Tubes in different groups were incubated at different temperatures. Gas pressure, $CH_4$, and $H_2$ were measured in different time intervals.

### Monitoring of the gas pressure, $CH_4$, and $H_2$

Gas pressures were recorded at different intervals using a pressure transducer (Honeywell 180PC Pressure Sensors, USA). $H_2$ and $CH_4$ were measured by a GC-TCD instrument (Agilent 7890B, USA) at those time-points. Gases were separated on packed GC columns (Porapak Q packing & MolSieve 5A packing, USA) at a column temperature of 80 °C, a 200 °C injection temperature, and a 200 °C TCD detector temperature; $N_2$ was the carrier gas. Amount (moles) of gases were calculated from the percentage of the gases on head-space and pressure using "Gas Laws ($PV = nRT$)".

### Scanning electron microscope

Cells of strains were collected at exponetial phase and rinsed with anaerobic diluting solution [101]. The supernatant was removed after centrifuging at 10,000*g* for 10 min. The procedure was repeated three times. And then the pellets were pretreated with 2.5% glutaraldehydesolution (with anaerobic diluting solution) and stored at 4 °C for more than 8 h. The fixed samples were washed three times with phosphate buffer, with each rinse lasting for 10 min. Subsequently, the samples were dehydrated using an ethanol gradient, starting with 50%, followed by 70, 80, 90, and 100%, each for 15 min (100% for 30 min, three times). The samples were then immersed in tert-butanol three times, each time for 30 min. Afterward, the samples were dried using a freeze-dryer. They were affixed to the sample stage with double-sided adhesive tape, with the observation side facing upward. A 10-nm gold coating was applied to the samples using an ion sputtering apparatus. Finally, the samples were prepared for scanning electron microscope (SEM) (S-3000N, HITACHI, Japan).

## Whole-genome sequencing and assembly

The genomes of the two strains were sequenced and assembled using the same procedure. The mixed solution in the logarithmic growth phase of the strains was centrifuged at 9000*g* for 10 min to remove the supernatant, and the cell pellets were collected after three washes with sterile double-distilled water. Genomic DNA was extracted from the cell pellets using a DNA Kit (E.Z.N.A.® Bacterial DNA Kit, Omega Biotek, USA).

The concentration, quality, and integrity of DNA were determined using a Nanodrop ND-1000 (Thermo Scientific, USA) and 0.8% agarose gel electrophoresis. Genome sequencing of the strains was conducted using the Illumina HiSeq 4000 platform and PacBio RS II platform. Libraries were constructed from high-quality DNA following the manufacturer's instructions for the different platforms. Illumina sequencing used 270 bp read libraries followed by 150 bp paired-end sequencing. Trimmomatic [80] (v.0.33) trimmed low-quality reads and adapters. Four single-molecule real-time (SMRT) cell zero-mode waveguide arrays generated PacBio sequencing subreads. Subreads of less than 1 kb were removed and corrected using Pbdagcon (https://github.com/jgurtowski/pbdagcon_python). All uncontested groups of genomic fragments were assembled using Canu [102] (v.2.2) against a high-quality circular consensus sequence (CCS) subreads set. Single-base sequence variations were corrected using GATK (v.4.2.2.0; https://github.com/broadinstitute/gatk). Illumina reads were mapped to a plasmid database (http://www.ebi.ac.uk/genomes/plasmid.html) using SOAP [103] (v.2.1.0) to identify plasmids in the genome.

### Genomic annotation and metabolic reconstruction

Gene calling and annotations for each genome from the different sources were performed using a standardized approach based on Prokka [104] (v.1.14.6) to ensure the reliability of subsequent analyses, which coordinates a range of existing external tools for selection. ARAGORN [105] (v.1.2.41) and Barrnap (v.0.9; https://github.com/tseemann/barrnap) were used to identify tRNAs and rRNAs. The MinCED [106] (v.0.4.2) program was performed to search for CRISPR throughout the genome. Protein coding sequences of each genome were predicted using Prodigal [107] (v.2.6.3) with the "-p single" option and then annotated using a combination of BLASTP (v.2.13.0) and HMMER [108] (v.3.3.2) aligned against the UniProt protein database. PhiSpy [109] (v.4.2.21) was used to identify prophages in the genome, and IMG/VR [110] (v.3), the largest publicly available viral sequence database, was used to identify the novelty of the virus. Functional classification and metabolic reconstruction were conducted by querying the predicted protein sequences against specific databases, including the eggNOG database using DIAMOND [111] (v.2.0.13; $E$-values $< 1e^{-5}$) based on BLASTP search, KEGG database using KofamScan [112] (v.1.1.0; high-scoring assignments with asterisks were considered reliable), and CAZyme database (v.7; http://www.cazy.org/) using HMMER [108] (v.3.3.2). Metabolic reconstruction was performed based on the KEGG-annotated pathways, and corresponding KOs across different genomes were integrated and summarized using an in-house developed script.

### Phylogenetic and phylogenomic analyses

A set of 26 ribosomal proteins [35, 113] and 118 archaeal marker proteins [36] were downloaded from the Pfam [114] (v.35.0) and TIGRFAM [115] (v.15.0) databases according to previous studies that covered a wide range of bacterial and archaeal diversity. The phylogenetic trees constructed based on these two marker proteins followed a shared process. The genomes were first scanned for these marker proteins using the HMMSEARCH program on HMMER [108] (v.3.3.2), and the best matched sequences were extracted. Each protein sequence from the genomes was aligned using MAFFT

[116] (v.7.487; parameter: -ep 0 -genafpair -maxiterate 1000) and filtered with TrimAL [117] (v.1.4.1; parameter: -automated1). Then, the marker protein sequences were concatenated into a single alignment using PhyloSuite [118] (v.1.2.2), and the maximum likelihood trees were built using IQ-Tree [119] (v.2.0.6) with the best-fit model GTR+F+R10 and 1000 ultrafast bootstrapping iterations. All trees produced were visualized and beautified in the online tool iTOL [120] (v.6; https://itol.embl.de/). The BtuC phylogenetic tree was constructed based on 281 aligned positions from the alignments of 109 protein sequences from Methanomassiliicoccales, 408 from other orders, and 245 from the NCBI-nr database using a model LG+F+R10. The result of gene prediction first extracted the 16S rRNA gene on each genome, and BLASTN (v.2.13.0) was used to align the SILVA rRNA database [121] (v.138.1) to identify the 16S rRNA gene manually. All 16S rRNA sequences were then integrated to build the tree using a model GTR+F+R4, following the same steps as before. Likewise, the *mcrA* genes from 199 Methanomassiliicoccales genomes and 18 Methanosarcinales genomes (outgroup taxa) were extracted and used to construct a maximum likelihood tree based on the LG+F+I+I+R4 model.

### Genome-relatedness calculations

Using the following tools to distinguish taxonomic levels, we calculated an overall genome-relatedness index from all possible pairwise comparisons. ANI based on either BLASTN (v.2.13.0; $ANI_b$) or MUMmer [122] (v.3.0; $ANI_m$) as alignment algorithms were implemented using pyANI [123] (v.0.2.12). AAI was calculated using the default parameters of the aai_wf program in CompareM (v.0.1.2; https://github.com/dparks1134/CompareM; $AAI_{cm}$). The 16S rRNA gene similarity of the two genomes was estimated using pairwise BLASTN (v.2.13.0) alignment. The taxonomic assignment of Methanomassiliicoccales was redefined based on the corresponding AAI values for two representative genomes; >45, >65, and >95% were identical families, genus, and species, respectively. The ANI value was also used to classify the same species (set at 95%). These standards followed the recommended descriptions [124, 125] to extend the categorization of uncultivated bacteria and archaea. The overall genome-relatedness matrices resulting from these analyses are presented in Additional file 4: Table S3.

### Comparative genomics

Orthologous genes were identified for comparative analysis of Methanomassiliicoccales using OrthoFinder [126] (v.2.5.4), a protein sequence search program based on DIAMOND [111] (v.2.0.13). Ancestral family genes were determined using the program COUNT [127] (v.9.1106) with Dollo parsimony, which strictly prohibited multiple gene gains and permitted reconstructing gene gain and loss events in observed species and potential ancestors. The gene gain events for various genera were calculated based on phylogenetic relationships and the presence or absence of genes within each genus clade. An in-house script generated gene distribution tables for pan-genome analysis of taxonomic genera based on gene presence or absence matrix. Genome alignments and locally collinear block identifications were performed using the progressive MAUVE plugin in Geneious [100] (v.2022.1.1). The genetic differences between LGM-DZ1 and the other 13 clustered MAGs were visualized using BRIG [128] (v.0.95).

**Analysis of the LME and HME rumen microbiome datasets**

The metagenomic and metatranscriptomic data for the low and high methane samples conducted by Shi et al. were retrieved from the NCBI SRA database [53]. Based on this study, methane measurements were conducted on 22 sheep and sorted by their mean production values. Four high emitters and four low emitters (each sampled at two time-points) were selected for further analysis. Metagenomic reads of 16 rumen samples from the LME and HME sheep were quality controlled, assembled, and binned using the consistent pipeline described above. A phylogenetic tree of four MAGs from this dataset and other ruminant genomes was constructed using PhyloPhlAn [129] (v.3.0.2). Contigs were combined, and genes were predicted using Prodigal [107] (v.2.6.3) with the "-p meta" option. CD-HIT [130] (v.4.8.1; parameter: -n 9 -g 1 -c 0.95 -G 0 -M 0 -d 0 -aS 0.9) clustered genes over 100 bp in length. The non-redundant gene catalog was taxonomically and functionally assigned using DIAMOND [111] (v.2.0.13; *E*-values $< 1e^{-5}$) based on BLASTP search and KofamScan [112] (v.1.1.0) against the NCBI-nr and KEGG databases. The corresponding metatranscriptomic reads were pre-processed using Fastp [131] (v.0.20.1; parameter: -detect_adapter_for_pe -q 25 -5 -3 -l 100) and mapped to the sheep genome (*Ovis aries*, GCA_002742125.1) using BWA-MEM [81] (v.0.7.17) to eliminate host RNA sequence. The ribosomal RNAs were filtered and removed using SortMeRNA [132] (v.4.2.0) against the indexed SILVA rRNA database [121] (v.138.1). Finally, messenger RNA reads from each group of eight sheep rumen samples were mapped to nucleotide sequences of the gene catalog from metagenomic data using BWA-MEM [81] (v.0.7.17), and gene expression abundance was calculated using the TPM (transcripts per million) algorithm. The TPM value for a gene represents the estimated number of transcripts per million reads in the sample, considering gene length and total sequencing depth [133]. Total gene expression for each genome in each sample was calculated using metaWRAP [134] (v.1.3) with a "quant_bins" module for TPM calculation.

**Statistical analyses**

The comparison of ancestral gene families in genera from the environmental and gastrointestinal clades, as well as the assessment of taxonomic and gene abundance differences between the LME and HME groups, were calculated using the Wilcoxon rank-sum test by the wilcox.test module in R (v.4.1.3) with the parameter "paired = FALSE," and the random forest for differential gene selection was constructed with 1000 trees using the R package randomForest (v.4.7-1; https://cran.r-project.org/web/packages/randomForest). PCoA plot of 192 Euryarchaeota and Crenarchaeota genomes, as well as 243 Methanomassiliicoccales genomes, revealed the genomic relationships using the Bray-Curtis dissimilarity matrix based on the profiles of gene orthologs, and the ANOSIM test in the R package vegan [135] (v.2.5-6) with 9999 permutations validated the differences. The correlations between Methanomassiliicoccales and other orders were calculated using the Hmisc module of R based on the Spearman correlation test with an asymptotic measure-specific *P* value.

## Supplementary Information

---

**Additional file 1: Supplementary Figures.** This additional file contains the supplementary figures (Figs. S1-S6).

**Additional file 2: Table S1.** Information on 146 representative genomes from 13 orders of Euryarchaeota and 46 outgroup genomes from phylum Crenarchaeota.

**Additional file 3: Table S2.** Information on top 31 significant differential genes defined by random forest classifier.

**Additional file 4: Table S3.** Information on 243 Methanomassiliicoccales genomes, five outgroup taxa from the order Thermoplasmatales, and 17 high-quality archaeal MAGs recovered from the LME and HME sheep included in this study.

**Additional file 5: Table S4.** Information on 12,093 orthologous genes inferred from 243 Methanomassiliicoccales genomes.

**Additional file 6: Table S5.** Information on 1656 orthologous genes from the common ancestor of Methanomassiliicoccales.

**Additional file 7: Table S6.** Information on 968 gained genes specific to 22 genera.

**Additional file 8: Table S7.** The central metabolic pathways shared by Methanomassiliicoccales.

**Additional file 9: Table S8.** Comparisons of genomic properties between LGM-RCC1 and LGM-DZ1.

**Additional file 10: Table S9.** The expression profiles of each genome in the LME and HME sheep.

**Additional file 11.** Review History

---

### Acknowledgements

We acknowledge the support of the high-performance computing platform of Bioinformatics Center, Nanjing Agricultural University. The authors would like to thank Prof. Le Luo Guan (Department of Agricultural, Food and Nutritional Science, University of Alberta, Edmonton, Canada) for helping to revise the manuscript.

### Review history

The review history is available as Additional file 11.

### Authors' contributions

S.M. and W.J. conceived, designed, and supervised the project. S.Z., X.Z., Y.Z., Y.L., and F.X. collected samples and performed experiments. F.X. carried out bioinformatic analyses and drafted the paper. W.J., S.M., G.T.A., P.B.P., and W.Z. revised and contributed ideas on the paper. All authors read, edited, and approved the final manuscript.

### Funding

This research was funded by the National Natural Science Foundation of China (project no. 31872381, 32072755, and 32272896). P.B.P. is supported by the Novo Nordisk Foundation (project no. 0054575-SuPAcow).

### Availability of data and materials

The genomes of the two cultured strains in the present study have been deposited in the European Nucleotide Archive under study accession no. PRJNA915581 (CP115555 and CP115556). All metagenomic sequencing data generated and analyzed in the present study have been deposited in the European Nucleotide Archive under study accession no. PRJNA915582 and PRJNA657455. All the recovered metagenome-assembled genomes are available for bulk download in Figshare https://figshare.com/s/7ba66239b6ee98bc9094 [136]. The reference information for all public datasets [37–40] and genomes used in this study is summarized in Additional file 2: Table S1 and Additional file 4: Table S3.

## Declarations

### Ethics approval and consent to participate

All animal-specific procedures were approved and authorized by the Nanjing Agricultural University Institutional Animal Care and Use Committee (No. SYXK-2017–0027).

### Competing interests

The authors declare that they have no competing interests.

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

## 