## [**Additional file 11.** Review History · Genome Biology]

Review History

First round of review

Reviewer 1

Are you able to assess all statistics in the manuscript, including the appropriateness of statistical tests used? No, I do not feel adequately qualified to assess the statistics.

Comments to author:

This study analyzed 243 Methanomassiliicoccales genomes from natural habits and animal GITs, assembled 33 high-quality MAGs from ruminant animal GITs and cultured two Methanomassiliicoccales strains. The results in this manuscript add significantly to the knowledge of phylogenomic diversity of Methanomassiliicoccales. However, some revisions are necessary to improve the overall comprehensibility of the paper.

Major comments:

1. Some key findings are missing in the abstract section, for example vitamin B12 exchanges of Methanomassiliicoccales may lead to decrease methane emission in ruminants, and what exact functional role that Methanomassiliicoccales play in methane emissions from ruminants.
2. Fig. 1. evolutionary divergence time of the order Methanomassiliicoccales. The molecular dating for archaea is too speculative, considering the topology structure for archaea phylogenetic analysis is still not stable and root of archaea is still unknown. I am afraid this part will raise many confuses. How was the molecular clock calibrated? Cyanobacteria, one of the most important groups in divergence time analysis, is not included in the tree in Fig. 1A. Is it necessary to do divergence time analysis? What the numbers in parenthesis representing Fig. 1A? Why you choose BtuC for the phylogeny in Fig. 1D?
3. The narrative structure of the introduction could be clearer. You summarized much about other methanogen lineages. In contrast, I suggest to focus on Methanomassiliicoccales. The scientific questions showing valuable methane mitigation should be raised.
4. L132 and Fig. 1D, which two genomes of Methanomassiliicoccales did author choose for amino acid comparison, and is this a shared pattern in all Methanomassiliicoccales and Methanosarcinales genomes?
5. L377, why the authors did not use obligate hydrogenotrophic MAGs that obtained from metagenome assembly and binning, but use seven public ruminant-derived genomes of Methanobrevibacter for metatranscriptomic analyses. I assumed that Methanobrevibacter the predominant obligate hydrogenotrophic methanogen in sheep ruminants, but the authors need to provide the type of obligate hydrogenotrophic methanogen, as well as the relative abundances of Methanomassiliicoccales and obligate hydrogenotrophic methanogen in ruminants of LME and HME sheep.
6. L377, according to Table S9, 67 Methanomassiliicoccales genomes were chosen for metatranscriptomic analyses. The authors need to explain the reason for selecting those genomes. In addition, why four MAGs recovered from sheep metagenomes were not included?

7. L388, do the genes *cbiM* and *cbiQ* belong to Methanomassiliicoccales? The authors did not mention these genes for B12 biosynthesis in the section of Methanomassiliicoccales genome reconstruction, and need to add the expression patterns of B12 biosynthesis and transport genes shown in Fig.4 and S5, at least *btuFCD*, to support the hypothesis that B12 exchanges between microbes may be an important driver allowing Methanomassiliicoccales to occupy specific niches.

8. Fig. 2. The order Methanomassiliicoccales is made up of two large clades: Environmental clade and Gastrointestinal clade. In Fig. 2A, we can see biome types are classified into three types: Host-associated, Anthropogenic and Environmental. Why there is inconsistency? Did you analyze phylogenomic diversity of Methanomassiliicoccales based on *mcrA* gene since it is an important marker gene?

9. Fig. 3. Are the number of gene gain events averages for different genera in Fig. 3B? What are specific genes in Fig. 3D? Why only retain 8 genera at the bottom of Fig.3D?

10. Fig. 5. Please complete the cultivation conditions other than the variables in Fig. 5C. For example, what is the pH and temperature in top left panel under different nutrients? Is the phylogenetic tree of G.21 or G.22 in Fig. 5E. There is inconsistency between figure legend and figure.

11. Fig. 6. Four assembled MAGs from LME and HME sheep are assigned into G. 21. Why do you compare genomic expression of 10 genomes from G. 16, G. 21, and G. 22 in Fig. 6C? What about other Methanomassiliicoccales genera? Are there any other related genes vary between LME and HME sheep groups except for *cbi*? It is quite interesting to propose strategies against methane emissions by investigating interactions between Methanomassiliicoccales and other microbes in the rumen. Do you find some microbes showing significant interactions with Methanomassiliicoccales? Fig.6C, what the light grey panel on the bottom left represented?

Minor comments:

L50: H₂/formate?

L78: Researches...have...

L118: Fig. 1C

L240: Cu²⁺

L459 and 460, the authors mentioned "sediments" and "highly enriched sediments" in materials sections for DNA extraction of two isolates. It is very confusing, as I assumed authors have the pure culture of Methanomassiliicoccales, not enrichments.

L562: messenger RNA reads, not microbial RNA reads.

L590: MgSO₄·7H₂O, CaCl₂·2H₂O

L599: D-Ca-pantothenate

L624: Add references for using lumazine as inhibitor for Methanobrevibacter. Does lumazine inhibit growth of Methanomassiliicoccales?

L626: How to make sure the isolate is pure, by PCR or just by microscopy? L654: H₂-dependent methylotrophic methanogenesis is not a fermentation process. Change the title to "monitoring of the gas pressure, CH₄ and H₂".

L663: (PV=nRT)".

L664: microscope.

L669: add details for preparation of microbiological specimens for SEM.

L1003 and Fig. 1: M. barkeri

Reviewer 2

Are you able to assess all statistics in the manuscript, including the appropriateness of statistical tests used? Yes, and I have assessed the statistics in my report.

Comments to author:

General Comments:

Abstract - Results presented in the abstract are more like conclusion. I would suggest authors to present results concisely. Similarly, conclusion is presented as a vague statement. With the absence of results I cannot see how authors presented strategies to mitigate methane emission by studying genomes of methanomassilicoccales

Introduction - Authors have presented a well-written background. However, I feel like this too skewed towards methane emission rather than phylogeny of Archaea. Authors can be straightforward and concise about lack of understanding on archaeal groups and why we need a deeper understanding of their genome and functionality. This way, the study will look more like a hypothesis-driven research rather than exploratory.

Results and Discussion - I would suggest authors to separate two sections. This is an extensive look into the genome and functionality. Therefore, authors have generated fundamental knowledge about methanomassilicoccales. But I feel like all that is buried under the overarching discussion. Also, it will help authors to present a critical yet concise discussion. The other thing I found hard is the lack of link to methods in the results. As the methods presented after results, I had to switch back and forth to really understand what is this result about.

Conclusion - I feel this a bit overarching approach to present conclusion. I would suggest authors to be straightforward with conclusions and only to minimize speculations or implications

Methods - sampling section seems to be over simplified. What is TMA? Did you culture only rumen archaea? How did you collect samples from other species? Did you collect fecal samples? how did you collect those fecal samples? What are the environmental samples that authors has been referring to throughout the manuscript? What are host databases used in removing host contaminations? Did you use metagenomics and metatranscriptomics data from a already published study? How did you decided LME and HME? Can the methods be presented in a better order? it goes from genome assembly to methane emission sheep trial to culturing. I got a bit lost as to where the rationale for moving from one to another. Statistical analysis needs details. What did authors compared using a non-parametric test

Figures - Legends needs to be bigger for all the plots, heatmaps, and schematic diagrams. Figure panels seems to be too crowded

Reviewer 3

Are you able to assess all statistics in the manuscript, including the appropriateness of statistical tests used? Yes, and I have assessed the statistics in my report.

Comments to author:

Unraveling the phylogenomic diversity of Methanomassiliicoccales: implications for mitigating ruminant methane emissions.

The widespread distribution of Methanomassiliicoccales, an order among Euryarchaeota that has the functionality to utilize "methyl" compounds (methyl amines and methanol) has created a renewed interest among scientific communities that work around anerobic microbial communities inhabiting host/habitats over the past decade. Several papers have discussed that the "rumen environment" is no less and has a rich community of Methanomassiliicoccales (Janssen and kirs, 2008, Poulsen et al., 2013, Borell et al., 2014, Sollinger et al., 2016, 2018 and 2019, Seedorf et al., 2015 and several others. Many of these have performed detailed comparisons of Methanomassiliicoccales distribution among the gastrointestinal tract including rumen to those of other environments. These studies highlighted the significance of adaptation of these methylotrophs to respective environments. Compared to these studies, the current study of Xie et al., is not completely novel, but an extension of the studies reported thus far. However, the authors have employed a robust analytical approach integrating multi programs and bioinformatic tools to integrate multiple metagenomic datasets from diverse environments, assemble MAGs to better understand evolutionary, phylogenetic, and functional roles of Methanomassiliicoccales with emphasis on the rumen environment. While the strength of the study is in the use of these sophisticated tools to provide new information on novel genera, genes and associations between pathways that have evolutionary significance, the study was constrained to link the findings to methane abatement.

Overall, the manuscript is well written and organized into sections that are summarized well and supported with graphic representations. Because discussion is integrated within results sections, the reasoning and justification of why the authors had to perform these complicated analyses is seriously lacking. This reviewer still sees the sections as discrete but not cohesive. A major

limitation I see in this study is the lack of hypotheses for each approach applied, take home message, strengths and limitations associated with each study section, and how the work is linked to methane mitigation.

General comments:

The first comment is that except for using metagenomic datasets, I see this study to be very similar to the efforts of Sollinger et al., 2016, and 2019. Comparisons clearly state that the diversity of Methanomassiliicoccales in the rumen or GI tracts and other environments are different. Identification of a few lineages as genera has also been discussed in Sollinger and Seedorf studies. Both 16S rDNA and McrA genes have been used to compare the novel genera in previous studies, whereas the current study used only 16S rDNA.

Second, the functional significance of Methanomassiliicoccales in the rumen has not been discussed. While an attempt to show the distribution of methyl amines, and methanol along with cofactor metabolism in B12 synthesis have been identified to enable Methanomassiliicoccales to cluster separately, the authors did not discuss the functional relevance. Vitamin B12 is part of corronoid proteins, the central core of methyltransferases and so discussion on methyl transferases and how these are distinct among different clades of Methanomassiliicoccales may have set a new direction for this study.

The third comment is while the authors have attempted to compare with Methanobrevibacter, why has there not been any attempts to compare this Methanomassiliicoccales with Methanosphaera? The latter methanogen belongs to Methanobacteriales, adapted to using methanol as the major substrate by having mtaA, mtaB and MtaC genes. Just curious why the study did not pick up the presence of MtaB and MtaC from Methanosphaera? Among methylotrophic methanogens, both Methanosphaera and Methanomassiliicoccales are considered for discussion. This paper may benefit from adding a section on comparison of Methanosphaera and Methanomassiliicoccales.

Fourth, the inclusion of metagenomic and metatranscriptomic data of high and low methane emitting sheep remains questionable. The authors have used metagenomic datasets from diverse ruminant species and described the distribution of two distinct clades: LGM-RCC1 and LGM-DZ1. None of these were identified in high or low methane emitting sheep. The genus identified was G.21 across all sheep. Except for the fact that Methanobrevibacter was enriched in HME compared to LME, whereas Methanomassiliicoccales between LME and HME sheep has not been justified very well. Shi et al 2014 described that LME may be enriched in Methanosphaera but there has not been an attempt to discuss connection between Methanosphaera and Methanomassiliicoclaes.

Finally, the title of the paper is to link Methanomassiliicoccales and implications to methane mitigation. Results and discussion around methanogenesis pathways, particularly distribution of genes coding for MCR and HDR enzymes, the key enzymes in methanogenesis has been poorly described. Shi et al 2014 highlights the distribution of mcr/mrt genes as critical for HME and LME and that mrt genes (MCR isoenzyme II) may be enriched in Methanomassiliicoccales and Methanosphaera, and in general in Methylotrophic methanogens. A comparison of mcr and mrt between different MAGs may add new insights on the distribution of Methanomassiliicoccales. Further, the use of Rice cluster C, the distribution of methanomassiliicoccales with and without

cytochromes, and the emergence of methanomassiliicoccales without cytochromes as evolutionary younger than with cytochromes, as described in Thauer et al 2008 has not been discussed.

The study is aimed to address an important topic that has been daunting to the ruminant systems, is to better understand the role of Methanomassiliicoccales. As discussed above, several papers have been published explaining the distribution of this relatively new clade of methanogens. While the study has invested significant efforts in assembly and comparisons, the underlying scientific description is rather weak, and will certainly benefit from revising the scientific content.

A good description of hypotheses with rationale for section of analysis, followed by a separate section of discussion, highlighting the connection between the findings and methanogenesis pathways, strengths and directions for the future are all needed to make this study more suitable for publication in Genome Biology, and to use this information for mitigation of enteric methane emissions from diverse ruminant systems.

Specific comments:

Abstract: Ln 18-20: Phylogeny and ecological roles have been described in previous reports (Thauer et al 2008; Sollinger et al 2015, 2019, Seedorf et al). The authors should mention these studies, and describe what the authors wanted to accomplish through this study.

Ln 21-25 has been done previously as detailed in the previous reports.

Ln 27-29: Not clear. Why wasn't the ISO4 strain isolated from sheep not used in analysis.

Ln 29-30: Vague statement.

Ln 31-33: If the functional role of Methanomassiliicoccales was proposed from HME and LME only, why was the analysis done to assemble MAGs? What is the connection between the Ln 93 to 365 and HME/LME analysis?

Ln 34-36: Conclusion is not well justified.

Introduction: This section requires major revision. The description of Methanomassiliicoccales in diverse environments and the rumen is very shallow. Several papers have been published attempting to understand the role of this clade. Discussion of those papers is needed.

Ln 67-68: The role of Methanomassiliicoccales in methane emissions is still speculative.

Ln 70-74: Link to TMAO is left hanging and is not connected with the rest of the section.

Hypotheses and objective(s) must be stated clearly and how the authors planned to accomplish.

Results and Discussion

Ln 92-109: Why was this analysis done?

Ln 99-102: sentence must be revised.

Ln:107-109: with and without cytochromes may explain their evolution as described in Thauer et al 2008.

Ln 110-148: what about Methanospaera? Although Methanobacteriales, it does have the mtaA, mtaB and mtaC and vit B12. Why was this not described? Also what is the involvement of cytochromes if it is related to evolution?

Ln 170-196: This section is interesting and provides insights on distribution across diverse environments. What genera were abundant among ruminant datasets?

Ln 197-247: This is a good exercise but the use of this information for cellular metabolism and methane emission is needed.

Ln 248-292: Variations based on methyltransferases, mcr/mrt and HdrABC and HdrDE may provide better insights to link the information to methanogenesis.

Ln LGM-DZI survives in acidic environment (abomasum). Are these detected in the ruminal environment?

Ln 366-396: This section on HME/LME appears disconnected with the remaining text.

Ln 404-406: The statement is not well supported with any data.

Ln 408: The role of Methanobrevibacter in high methane emissions has already been reported in Shi et al and does not connect with this study.

Ln 415-475: The authors have done multiple steps in collecting natural digesta samples from different ruminant species, performed enrichment and followed by metagenomic analysis. This entire work was not emphasized nowhere in the introduction or results/discussion. This reviewer was confused between what was downloaded from public databases vs. what has been generated in-house. Coming back to my general comment, the authors could have generated multiple hypotheses from the type of work done. Culture vs. metagenomics done in house vs. publicly downloaded datasets vs. HME and LME sheep.

Better description of TMA enrichment and rationale for enrichment is needed. Why only TMA, and why not DMA and MMA and also methanol?

Ln 574-580: what is the need for rumen and abomasum contents? Methane is generated only from the rumen?

Were there any new isolates of methanogens?

Ln 655: It is often suggested that while TCD can detect methane, FID provides accurate measurements on methane emissions. A suggestion to use FID specifically using on methanogenic isolates.

Ln 664: typo

Figure 6: What is the unit TME stand for in the plot for bacterial and archaeal gene expression.

Figure 6C: In Shi et al. 4 HME and 4LME sheep were selected. In the current study, 8 animals each are selected. Is it possible to add methane emissions to each of these sheep as there appears to be a large variation in the distribution of Methanobrevibacter and RGM groups. Again it is surprising that Methanosphaera is not listed as Shi et al indicated that LME sheep have been enriched in Methanosphaera.

The point-by-point response to the comments

Reviewer #1

This study analyzed 243 Methanomassiliicoccales genomes from natural habits and animal GITs, assembled 33 high-quality MAGs from ruminant animal GITs and cultured two Methanomassiliicoccales strains. The results in this manuscript add significantly to the knowledge of phylogenomic diversity of Methanomassiliicoccales. However, some revisions are necessary to improve the overall comprehensibility of the paper.

R: We appreciate your time in reviewing our manuscript and providing these constructive comments. We believe these suggestions have helped to improve it.

Major comments:

Q1. Some key findings are missing in the abstract section, for example vitamin B₁₂ exchanges of Methanomassiliicoccales may lead to decrease methane emission in ruminants, and what exact functional role that Methanomassiliicoccales play in methane emissions from ruminants.

R: We agree with your suggestion and have made modifications in the abstract, incorporating key findings as follows *“Here, we profiled and analyzed 243 Methanomassiliicoccales genomes assembled from both cultured representatives and uncultured metagenomes recovered from various anthropogenic and natural biomes, including the GITs of different animal species. Our analyses reveal the presence of numerous undefined genera and suggest genetic variability in unique metabolic capabilities within Methanomassiliicoccales lineages, which is essential for adapting to their respective ecological niches. In particular, the GIT Methanomassiliicoccales demonstrate the presence of co-diversified members with their hosts over evolutionary timescales and likely originated in the natural environment. We highlight the presence of diverse clades of vitamin transporter BtuC proteins that distinguish Methanomassiliicoccales from other archaeal orders and likely provide them a competitive advantage in efficiently handling B₁₂. Furthermore, genome-centric metatranscriptomic analysis of ruminants with varying methane yields revealed elevated expression of select Methanomassiliicoccales genera in low methane animals and suggests that B₁₂ exchanges could crucially enable them to occupy ecological niches that possibly alter the direction of H₂ utilization.”* in lines 20-32.

Q2. Fig. 1. evolutionary divergence time of the order Methanomassiliicoccales. The molecular dating for archaea is too speculative, considering the topology structure for archaea phylogenetic analysis is still not stable and root of archaea is still unknown. I am afraid this part will raise many confuses. How was the molecular clock calibrated? Cyanobacteria, one of the most important groups in divergence time analysis, is not included in the tree in Fig. 1A. Is it necessary to do divergence time analysis? What the numbers in parenthesis representing Fig. 1A? Why you choose BtuC for the phylogeny in Fig. 1D?

R: Thank you for your comments. In this study, we calibrated the molecular clock using the recommended time nodes for Euryarchaeota and Crenarchaeota from the resource of TimeTree of Life (TTOL; <http://www.timetree.org/>), which has integrated the SSU sequences of 684 species of cyanobacteria from the NCBI database as crucial molecular time markers (Marin et al., 2017). We agree with the reviewer that the uncertainty regarding the root of archaea and this part is relatively weak. Thus, for accuracy and cautious, we have removed the divergence time analysis in the revised manuscript.

For Fig. 1A, we have added a description in the figure legend of Fig. 1A *“The numbers in the brackets represent the number of genomes corresponding to each order.”* in lines 1209-1210.

We chose BtuC due to that the membrane protein BtuC is capable of transporting vitamin B₁₂ into

the cell. Given the redundancy of BtuC genes within Methanomassiliicoccales, we would like to know if BtuC diversity may be associated with their environmental adaptability. In the revised manuscript, we have added an explanation “Research has confirmed that multiple transporters in *Bacteroides thetaiotaomicron* exhibit distinct preferences for corrinoids and contribute to microbial adaptability in the human gut [56]. To explore the diversity of BtuC in Methanomassiliicoccales, a protein tree was constructed based on BtuC homology from various public sources.” in lines 154-157 and a schematic to illustrate the function of the BtuC protein in Fig. 1D.

Q3. The narrative structure of the introduction could be clearer. You summarized much about other methanogen lineages. In contrast, I suggest to focus on Methanomassiliicoccales. The scientific questions showing valuable methane mitigation should be raised.

R: Thanks for this constructive suggestion. We have reorganized the Introduction section, emphasizing the research significance of Methanomassiliicoccales, discussing recent research developments related to Methanomassiliicoccales over the past decade, addressing research gaps concerning Methanomassiliicoccales in the rumen of ruminants, and outlining the objectives of this study (in lines 38-112).

Q4. L132 and Fig. 1D, which two genomes of Methanomassiliicoccales did author choose for amino acid comparison, and is this a shared pattern in all Methanomassiliicoccales and Methanosarcinales genomes?

R: Thank you for your comment. Here, we searched a total of eight complete genomes of Methanomassiliicoccales and selected the protein sequences from the two most similar genomes. As suggested by Reviewer #3, we have removed this analysis in the revised manuscript and instead focused on cytochromes. Fig. 1D has also been revised accordingly.

Q5. L377, why the authors did not use obligate hydrogenotrophic MAGs that obtained from metagenome assembly and binning, but use seven public ruminant-derived genomes of *Methanobrevibacter* for metatranscriptomic analyses. I assumed that *Methanobrevibacter* the predominant obligate hydrogenotrophic methanogen in sheep ruminants, but the authors need to provide the type of obligate hydrogenotrophic methanogen, as well as the relative abundances of Methanomassiliicoccales and obligate hydrogenotrophic methanogen in ruminants of LME and HME sheep.

R: In this study, we reanalyzed rumen metagenomic data from the LME and HME sheep, identifying 17 high-quality archaeal MAGs. These MAGs comprised one *Methanosphaera*, four Methanomassiliicoccales, and 12 *Methanobrevibacter* genomes. Consequently, we selected *Methanobrevibacter* as a control in our study for comparison with Methanomassiliicoccales. We did not directly choose MAGs obtained from the LME and HME samples for abundance calculations. This is because these MAGs were constructed from the assembly of metagenomic reads, and direct alignment could result in artificially inflated abundances from the source samples, potentially affecting the accuracy of abundance assessments. In other words, we might observe a MAG with exceptionally high abundance values in the sample from which it is generated, leading to an elevation in the average abundance or expression. To avoid such technical errors, we chose seven complete reference *Methanobrevibacter* genomes from ruminant GIT sources available on NCBI.

In the revision, we have included an analysis of methanogenic lineages at the order level according

to reviewer's comments, revealing a substantial increase in gene expression of Methanobacteriales, Methanomassiliicoccales, Methanomicrobiales, Methanococcales, and Methanonatronarchaeales in the HME sheep. We have added an explanation "...Further analysis of the methanogenic lineages at the order level showed a significant increase in gene expression of Methanobacteriales, Methanomassiliicoccales, Methanomicrobiales, Methanococcales, and Methanonatronarchaeales in the HME sheep (**Fig. 6A** and **Fig. S6B**). Focusing specifically on the Methanobacteria, Shi et al. revealed higher relative abundances of organisms belonging to *Methanobrevibacter* spp. in the HME sheep [53]. However, due to a lack of Methanomassiliicoccales genome representation at the time of the Shi et al. study, their transcriptional activity in the LME and HME animals was not recorded..." in lines 341-347.

Q6. L377, according to Table S9, 67 Methanomassiliicoccales genomes were chosen for metatranscriptomic analyses. The authors need to explain the reason for selecting those genomes. In addition, why four MAGs recovered from sheep metagenomes were not included?

R: For the metatranscriptomic "reanalysis", we employed a dataset consisting of 68 Methanomassiliicoccales and seven *Methanobrevibacter* genomes derived from the GITs of ruminants as our reference genome dataset. The Methanomassiliicoccales genomes from environmental sources were excluded, as we believe that these genomes derived from ruminants more accurately represent the abundance of Methanomassiliicoccales communities in the LME and HME sheep. In the revision, we have added an explanation "...Therefore, we further aligned transcriptomic reads from the LME and HME sheep to 68 ruminant-derived Methanomassiliicoccales genomes and seven publicly available genomes from the *Methanobrevibacter* genus..." in lines 347-349.

The four Methanomassiliicoccales MAGs were excluded to avoid technical errors, as explained in the previous response. In the revision, we have improved the analysis of these MAGs recovered from the LME and HME sheep "To advance this analysis further, we reanalyzed metagenomic data from both the LME and HME groups recovering 17 high-quality archaeal MAGs affiliated to *Methanosphaera* ($n = 1$), *Methanomassiliicoccales* ($n = 4$, classified as *Methanomethylophilus* in the GTDB) and *Methanobrevibacter* ($n = 12$) (**Table S3**). The phylogenomic positions of these four Methanomassiliicoccales MAGs maintained taxonomic consistency all clustering within G. 21 (**Fig. 6C**). In support of our hypothesis that G. 21 Methanomassiliicoccales populations are more active in the LME sheep, three of the four MAGs were obtained from the LME samples, signifying the dominance (**Fig. 6C**)." in lines 357-365.

Q7. L388, do the genes *cbiM* and *cbiQ* belong to Methanomassiliicoccales? The authors did not mention these genes for B₁₂ biosynthesis in the section of Methanomassiliicoccales genome reconstruction, and need to add the expression patterns of B₁₂ biosynthesis and transport genes shown in Fig.4 and S5, at least *btuFCD*, to support the hypothesis that B₁₂ exchanges between microbes may be an important driver allowing Methanomassiliicoccales to occupy specific niches.

R: Thank you for the suggestion. Yes, the cobalt transport system (*cbiOMQN*) was present in the Methanomassiliicoccales genomes (Fig. S5). In the revision, we have provided a description in the section of Methanomassiliicoccales genome reconstruction "We found that the genes involved in the anaerobic B₁₂ biosynthesis pathway synthesizing adenosylcobalamin from precorrin-2 and cobalt transport system (*cbiOMQN*) were largely encoded by Methanomassiliicoccales..." in lines 265-267.

According to the reviewer's suggestion, we have included the expression patterns of B₁₂ synthesis

and transport (*btuFCD*) genes in the LME and HME sheep to support our hypothesis. We found a significant increase in the total gene expression of multiple synthesis-related genes (*gltB*, *gltD*, *glnA*, *MET8*, *hemA*, and *cblL*) and the vital transporter BtuC (*btuC*, $P < 0.001$; Fig. 6D) in the rumen microbiome of the LME sheep. We have also added the results “*In light of this, we examined the B₁₂ biosynthesis pathway in the rumen microbiome and observed elevated levels of gene expression in multiple processes (gltB, gltD, glnA, MET8, hemA, and cblL) within the LME sheep (Fig. S6D). Furthermore, we also observed a significant increase in the total gene expression of the vital B₁₂ transporter BtuC (btuC, P < 0.001; Fig. 6D) in the rumen microbiome of the LME sheep.*” in lines 374-378 and the discussion “*...The increased expression of cobalt transporters in the LME sheep supports that a substantial number of components required for B₁₂ biosynthesis can be efficiently transported into microbial cells in the rumen when methane emissions are low. In correspondence, we also observed that multiple genes involved in the B₁₂ biosynthesis pathway in the rumen microbiome were upregulated in the LME sheep. Furthermore, the elevated gene expression of the BtuC transport protein in the LME sheep suggests that Methanomassiliicoccales-affiliated organisms with a higher demand for B₁₂ are more active under low-methane production conditions. B₁₂ interactions among rumen microbes could serve as a crucial mechanism, enabling specific members of Methanomassiliicoccales to inhabit ecological niches and thereby alter the direction of H₂. This points towards nutrient availability (such as B₁₂) playing an important role in regulating methane emissions in ruminants [59, 75-77].*” in lines 529-538.

Q8. Fig. 2. The order Methanomassiliicoccales is made up of two large clades: Environmental clade and Gastrointestinal clade. In Fig. 2A, we can see biome types are classified into three types: Host-associated, Anthropogenic and Environmental. Why there is inconsistency? Did you analyze phylogenomic diversity of Methanomassiliicoccales based on *mcrA* gene since it is an important marker gene?

R: We thank the reviewer for pointing to these important questions. In Fig. 2A, we classified the samples into three biome types based on their source, including “Host-associated”, “Anthropogenic”, and “Environmental”. The anthropogenic type primarily comprises samples from non-natural environments, such as reactor mud, fuel cells, and industrial wastes. In these samples, strains commonly utilized are isolates from the natural environments or the gastrointestinal tracts. Thus, in accordance with Fig. 2A, strains of anthropogenic type have been incorporated into both the environmental and gastrointestinal clades. To address this concern, we have added an explanation “*The strains isolated from anthropogenic samples (e.g., reactor mud, fuel cells, and industrial wastes) are mainly from the natural environment or the GITs of animals. Therefore, we observed that anthropogenic genomes have been incorporated into both environmental and gastrointestinal clades (Fig. 2A).*” in lines 175-178.

For classification, we used both the 16S rRNA gene tree and the whole-genome tree constructed from a concatenated set of 118 archaeal marker proteins in the phylogenomic analysis. We agree with the reviewer that the *mcrA* gene is an important marker gene for methanogens. As a result, we have incorporated phylogenomic analysis using the *mcrA* gene into the revised manuscript, and found that the taxonomy closely aligns with the whole-genome phylogenomic tree. We have added the results “*Maximum likelihood trees were constructed using 16S rRNA genes from 141 genomes, mcrA genes from 199 genomes and whole-genomes (n = 243), which collectively support the Methanomassiliicoccales order comprising of one family with significant variation at the genus level (Fig. 2B and Fig. S2).*” in lines 187-190 and the methods “*Likewise, the mcrA genes from 199*

Methanomassiliicoccales genomes and 18 Methanosarcinales genomes (outgroup taxa) were extracted and used to construct a maximum likelihood tree based on the LG+F+I+I+R4 model.” in lines 771-773.

Q9. Fig. 3. Are the number of gene gain events averages for different genera in Fig. 3B? What are specific genes in Fig. 3D? Why only retain 8 genera at the bottom of Fig.3D?

R: The gene gain events presented for different genera are not the averages of all genomes. In this study, gene gain events for various genera were calculated using the Count software (v.9.1106), relying on phylogenetic relationships and the presence or absence of genes within each genus clade. We have included a description for clarity in Materials and methods section of the revised manuscript “*The gene gain events for various genera were calculated based on phylogenetic relationships and the presence or absence of genes within each genus clade.*” in lines 791-793.

For Fig. 3D, the specific OGs represent a total of 968 gained genes that are unique to each genus. We have found that the majority of these genes are unclassified with respect to known protein functions, and only 8 genera have specific OGs with known functions. We have improved the description in the Figure legend as “*The number of unique genes specific to each genus among the 968 gained genes is shown at the top, and their KO annotations are shown at the bottom. The heatmap of the 40 annotated genes distributed across 8 genera is expanded to the right.*” in lines 1250-1252.

Q10. Fig. 5. Please complete the cultivation conditions other than the variables in Fig. 5C. For example, what is the pH and temperature in top left panel under different nutrients? Is the phylogenetic tree of G.21 or G.22 in Fig. 5E. There is inconsistency between figure legend and figure.

R: We have included the experimental conditions for Fig. 5C in the revised figure and corrected the description of Fig. 5E in line 1270 according to reviewer’s comments.

Q11. Fig. 6. Four assembled MAGs from LME and HME sheep are assigned into G. 21. Why do you compare genomic expression of 10 genomes from G. 16, G. 21, and G. 22 in Fig. 6C? What about other Methanomassiliicoccales genera? Are there any other related genes vary between LME and HME sheep groups except for *cbi*? It is quite interesting to propose strategies against methane emissions by investigating interactions between Methanomassiliicoccales and other microbes in the rumen. Do you find some microbes showing significant interactions with Methanomassiliicoccales? Fig.6C, what the light grey panel on the bottom left represented?

R: In this section, we identified 12 significantly differential genomes among the 68 Methanomassiliicoccales genomes derived from the GITs of ruminants. Moreover, these genomes were classified into G. 16, G. 21, and G. 22. As mentioned in the previous question, we have analyzed the expression patterns of B₁₂ synthesis and transport (*btuFCD*) genes in the revised manuscript and observed a significant increase in the total gene expression of multiple synthesis-related genes (*gltB*, *gltD*, *glnA*, *MET8*, *hemA*, and *cblL*), as well as the vital transporter BtuC (*btuC*, $P < 0.001$; Fig. 6D), in the rumen microbiome of the LME sheep.

In the revision, we have also provided the correlations between Methanomassiliicoccales and other microorganisms, and the results are as follows: “*We also identified significant correlations between Methanomassiliicoccales and other microorganisms at the order level (|Spearman’s correlation coefficient| > 0.8 and $P < 0.001$). This included 22 positive correlations, involving 15 bacterial orders, five fungal orders, one Ciliophora order, and one archaeal order, as well as three negatively correlated*

orders (**Fig. S6C**).” in lines 366-369. Additionally, we have found that *M. stadtmanae*, a H₂-dependent methylotrophic methanogens, exhibited a higher abundance and expression in the LME sheep. This may be driven by their similar metabolic characteristics in the utilization of methyl compounds. We have added the results “*Shi et al. previously reported a higher relative abundance of metabolically similar Methanosphaera spp. in the LME sheep [53]. However, at a transcriptomic level, we observed numerically higher gene expression levels in M. stadtmanae in the LME sheep, although the differences were not statistically significant (Fig. S6B).*” in lines 354-357 and the discussion “...Interestingly, *Shi et al. found that M. stadtmanae, which are also H₂-dependent methylotrophic methanogens, exhibited higher abundance in the LME sheep [53]; our re-analysis confirmed this observation although with no significant difference. The genera (G. 21, G. 22, and Methanosphaera spp.) with similar metabolic characteristics are likely to serve as indicative microbes for an LME microbiome and may be applied in future breeding programs for ruminants.*” in lines 520-524.

For Fig. 6C, the grey panel represents *Methanobrevibacter* genomes, and we have added annotations to the corrected figure (now at Fig. 6B).

Minor comments:

Q1. L50: H₂/formate?

R: Corrected as suggested.

Q2. L78: Researches...have...

R: Corrected as suggested.

Q3. L118: Fig. 1C

R: Corrected as suggested.

Q4. L240: Cu²⁺

R: Corrected as suggested.

Q5. L459 and 460, the authors mentioned "sediments" and "highly enriched sediments" in materials sections for DNA extraction of two isolates. It is very confusing, as I assumed authors have the pure culture of Methanomassiliicoccales, not enrichments.

R: Sorry for this ambiguity. We have replaced the terms “sediments” and “highly enriched sediments” with “cell pellets” to enhance clarity in lines 721-722.

Q6. L562: messenger RNA reads, not microbial RNA reads.

R: Corrected as suggested.

Q7. L590: MgSO₄·7H₂O, CaCl₂·2H₂O

R: Corrected as suggested.

Q8. L599: D-Ca-pantothenate

R: Corrected as suggested.

Q9. L624: Add references for using lumazine as inhibitor for *Methanobrevibacter*. Does lumazine

inhibit growth of Methanomassiliicoccales?

R: Thanks for your suggestion. Lumazine, as confirmed by Ungerfeld et al., is an effective inhibitor of pure cultures of ruminal methanogens, such as *Methanobrevibacter ruminantium*, *Methanosarcina mazei* and *Methanomicrobium mobile*. Padmanabha et al. have found that lumazine had no inhibitory effect on Methanomassiliicoccales strain. Therefore, we used lumazine to inhibit the growth of *Methanobrevibacter* spp. in the enrichment culture. We have included these two references in the revised manuscript in lines 654-655.

Q10. L626: How to make sure the isolate is pure, by PCR or just by microscopy?

R: In this study, the assessment of strain purification primarily involved amplifying the 16S rRNA gene for bacteria and methanogens, as well as conducting single-clone sequencing and alignment. We have incorporated detailed descriptions of these procedures in Materials and methods section of the revised manuscript “*The purification assessment of strains followed a two-step process. First, DNA was extracted from the enrichment culture as mentioned above. Universal primers were used for PCR amplification of the 16S rRNA genes of both bacteria and methanogens. For bacterial amplification, the 8F (5'-CACGGATCCAGAGTTTGAT(C/T)(A/C) TGGCTCAG-3') and 1510R (5'-GTGAAGCTTACGG(C/F)TACCTTGTTACGACTT-3')* primers were employed [99]. Methanogen amplification was carried out using the 86F (5'-GCTCAGTAACACGTGG-3') and 1340R (5'-CGGTGTGTGCAAGGAG-3') primers [100]. Subsequently, agarose gel electrophoresis was employed to confirm the absence of bacterial contamination in the amplification products. Second, the PCR products were purified using a PCR product purification kit (Vazyme, China), followed by insertion into the pESI-T vector using a cloning kit (Tsingke, China). These constructs were then transformed into competent cells (*Escherichia coli* DH5 α) and cultured in LB medium. The transformed cells were spread onto LB agar supplemented with ampicillin and incubated at 37 °C for 10 h. Single colonies were selected for further cultivation in LB medium, and the resulting culture liquid was subjected to sequencing. The obtained sequencing results were compared and analyzed using Geneious [101] (v.2022.1.1) to ensure base consistency, with all sequences consistently identified as pure strains.” in lines 657-672.

Q11. L654: H₂-dependent methylotrophic methanogenesis is not a fermentation process. Change the title to "monitoring of the gas pressure, CH₄ and H₂".

R: Corrected as suggested.

Q12. L663: (PV=nRT)".

R: Corrected as suggested.

Q13. L664: microscope.

R: Corrected as suggested.

Q14. L669: add details for preparation of microbiological specimens for SEM.

R: As suggested, we have included a more detailed description of the SEM in the revised manuscript “*And then the pellets were pretreated with 2.5% glutaraldehydesolution (with anaerobic diluting solution), and stored at 4 °C for more than 8 h. The fixed samples were washed three times with phosphate buffer, with each rinse lasting for 10 min. Subsequently, the samples were dehydrated using*

an ethanol gradient, starting with 50%, followed by 70%, 80%, 90%, and 100%, each for 15 min (100% for 30 min, three times). The samples were then immersed in tert-butanol three times, each time for 30 min. Afterward, the samples were dried using a freeze-dryer. They were affixed to the sample stage with double-sided adhesive tape, with the observation side facing upward. A 10 nm gold coating was applied to the samples using an ion sputtering apparatus. Finally, the samples were prepared for scanning electron microscope (SEM) (S-3000N, HITACHI, Japan). ” in lines 709-717.

Q15. L1003 and Fig. 1: M. barkeri

R: Corrected as suggested.

Reviewer #2

General Comments:

Q1. Abstract - Results presented in the abstract are more like conclusion. I would suggest authors to present results concisely. Similarly, conclusion is presented as a vague statement. With the absence of results, I cannot see how authors presented strategies to mitigate methane emission by studying genomes of methanomassiliicocales.

R: Thanks for your suggestion. We have revised the abstract and provided more results of this study as follows *“Here, we profiled and analyzed 243 Methanomassiliicocales genomes assembled from both cultured representatives and uncultured metagenomes recovered from various anthropogenic and natural biomes, including the GITs of different animal species. Our analyses reveal the presence of numerous undefined genera and suggest genetic variability in unique metabolic capabilities within Methanomassiliicocales lineages, which is essential for adapting to their respective ecological niches. In particular, the GIT Methanomassiliicocales demonstrate the presence of co-diversified members with their hosts over evolutionary timescales and likely originated in the natural environment. We highlight the presence of diverse clades of vitamin transporter BtuC proteins that distinguish Methanomassiliicocales from other archaeal orders and likely provide them a competitive advantage in efficiently handling B₁₂. Furthermore, genome-centric metatranscriptomic analysis of ruminants with varying methane yields revealed elevated expression of select Methanomassiliicocales genera in low methane animals and suggests that B₁₂ exchanges could crucially enable them to occupy ecological niches that possibly alter the direction of H₂ utilization.”* in lines 20-32.

Q2. Introduction - Authors have presented a well-written background. However, I feel like this too skewed towards methane emission rather than phylogeny of Archaea. Authors can be straightforward and concise about lack of understanding on archaeal groups and why we need a deeper understanding of their genome and functionality. This way, the study will look more like a hypothesis-driven research rather than exploratory.

R: In response to this comment and a similar one made by Reviewer #1, we have revised the Introduction section as *“The genomes of these cultured methanogens have yielded fundamental insights into the biology of Methanomassiliicocales [26, 29-32]. Nevertheless, research on members of the order Methanomassiliicocales has progressed slowly because of difficulties in isolation and cultivation, and constraints to study them under strict anaerobic conditions [33]. The phylogenetic taxonomy, functional characterization, and metabolic roles of Methanomassiliicocales remain inadequately characterized and have been restricted to a few isolated strains and genomes [26, 30, 34]. This is despite recent transformations in metagenomic technologies that have facilitated hundreds of thousands of metagenome-assembled genomes (MAGs) being generated from diverse habitats, including numerous archaeal methanogens [35-41]. Genomic analyses of selected MAGs in several studies have revealed that many previously uncultured lineages are indeed H₂-dependent methylotrophic methanogens and have initiated efforts to assess the diversity of taxa that are challenging to cultivate [3, 42, 43].”* in lines 74-84.

Additionally, we have outlined the specific hypotheses and research questions that this study as *“Here, we collected 208 publicly available Methanomassiliicocales genomes and MAGs from various global habitats as well as reconstructed 33 high-quality Methanomassiliicocales-affiliated MAGs. MAGs were recovered from 370 metagenomic datasets that were generated across 10 GIT regions of seven different ruminant species and eight metagenomic samples enriched with trimethylamine from*

cow rumen fluids. On our newly acquired genome atlas, we applied a large-scale genome-resolved comparison to reveal the essential metabolic functions of Methanomassiliicoccales and clarify the evolutionary classification of Methanomassiliicoccales originating from the GITs of ruminants. Subsequently, a genome-centric isolation strategy and in vitro enrichments of samples collected from the goat rumen and cow abomasum obtained pure culture strains of the major Methanomassiliicoccales groups. Finally, we additionally integrated metagenomic and metatranscriptomic data from Shi et al. [53] to provide novel insights into the distinct Methanomassiliicoccales groups in the rumen and their roles in methanogenesis.” in lines 101-112.

Q3. Results and Discussion -I would suggest authors to separate two sections. This is an extensive look into the genome and functionality. Therefore, authors have generated fundamental knowledge about methanomassiliicoccales. But I feel like all that is buried under the overarching discussion. Also, it will help authors to present a critical yet concise discussion. The other thing I found hard is the lack of link to methods in the results. As the methods presented after results, I had to switch back and forth to really understand what is this result about.

R: According to reviewer’s suggestion, we have separated the results and discussion sections, added more explanations in the results, and provided more detailed methods and reordering them in the revised manuscript. We believe that this significantly enhances the comprehensibility and fluency of the manuscript.

Q4. Conclusion -I feel this a bit overarching approach to present conclusion. I would suggest authors to be straightforward with conclusions and only to minimize speculations or implications.

R: In response to these comments and those made by Reviewers #1 and #3, we have made changes to this section by presenting it more concisely and incorporating more key results as “*Our study indicates numerous undefined lineages within Methanomassiliicoccales primarily detected through current uncultured MAGs, highlighting their substantial phylogenetic diversity. Genetic variability among Methanomassiliicoccales genera has led to functional partitioning and ecological divergence, resulting in distinctive metabolic capabilities crucial for their respective ecological niches. The gastrointestinal clades of Methanomassiliicoccales have demonstrated the potential for fidelity with their hosts over evolutionary timescales and likely originated in the natural environment. The central metabolic pathways shared by cultured genomes and uncultured MAGs of Methanomassiliicoccales revealed that B₁₂ is a crucial cofactor for Methanomassiliicoccales in the utilization of methyl compounds during methanogenesis. Additionally, an integrated genome-centric metatranscriptomic analysis revealed that several genera of Methanomassiliicoccales were more transcriptionally active in low-emitting animals, where elevated gene expressions of B₁₂ synthesis and transport may be associated with greater B₁₂ sharing in the LME sheep. Overall, these findings provide novel insights into the evolutionary history, metabolic cycling, and community function of the significant methanogenic order Methanomassiliicoccales.*” in lines 540-553.

Q5. Methods - sampling section seems to be over simplified. What is TMA? Did you culture only rumen archaea? How did you collect samples from other species? Did you collect fecal samples? how did you collect those fecal samples? What are the environmental samples that authors has been referring to throughout the manuscript? What are host databases used in removing host contaminations? Did you use metagenomics and metatranscriptomics data from a already published study? How did you decided

LME and HME? Can the methods be presented in a better order? it goes from genome assembly to methane emission sheep trial to culturing. I got a bit lost as to where the rationale for moving from one to another. Statistical analysis needs details. What did authors compared using a non-parametric test.

R: Thanks for reviewer's valuable comments and suggestions. We have revised Methods sections to include more details.

First, we have added the full name of TMA (trimethylamine). In this study, we slaughtered seven different ruminant species, collecting digesta samples from 10 regions within their gastrointestinal tracts for metagenomic sequencing to obtain the genomes of Methanomassiliicoccales strains in their natural states. Moreover, we utilized gastrointestinal digesta samples from these seven ruminant animals for enriching and isolating Methanomassiliicoccales strains. Additionally, genomes from humans, non-ruminant animals, and the global environment were retrieved from publicly available databases. Environmental samples encompassed a range of sources, such as soil, peat, wastewater, lakes, seas, hot springs, mangrove sediments, and groundwater. The host database was created using a collection of the closest animal genomes obtained from the NCBI database. We have included a description as "*The gastrointestinal digesta samples from the seven ruminant animals, including dairy cattle, water buffalo, yak, goat, sheep, roe deer, and water deer, were used for enriching and isolating Methanomassiliicoccales strains.*" in lines 607-609, "*Other genomes from humans, non-ruminant animals, and the global environment, including those from the Earth's Microbiomes project [38], were retrieved from the NCBI publicly available database (<https://www.ncbi.nlm.nih.gov/>) or links provided by the authors. Reference and genomic information for all the Methanomassiliicoccales genomes used in this study are summarized in **Table S3.***" in lines 601-605, and "*Specifically, we used the genome sets of hosts, including dairy cattle (*Bos Taurus*, GCA_002263795.2), water buffalo (*Bubalus bubalis*, GCA_003121395.1), yak (*Bos mutus*, GCA_000298355.1), goat (*Capra hircus*, GCA_001704415.1), sheep (*Ovis aries*, GCA_002742125.1), roe deer (*Capreolus pygargus*, GCA_000751575.1) and water deer (*Hydropotes inermis*, GCA_006459105.1); plants, including sorghum (*Sorghum bicolor*, GCA_000003195.3), wheat (*Triticum aestivum*, GCA_002220415.3), robur (*Quercus robur*, GCA_900291515.1), sweet potato (*Ipomoea batatas*, GCA_002525835.2), medicago (*Medicago truncatula*, GCA_000219495.2), rice (*Oryza sativa*, GCF_000005425.2), barley (*Hordeum vulgare*, GCA_900075435.2), maize (*Zea mays*, GCA_003185045.1 and GCA_000005005.6), soybean (*Glycine max*, GCA_000004515.4), and ryegrass (*Lolium perenne*, GCA_001735685.1); and human (*Homo sapiens*, GCA_000001405.28), as references for mapping to decrease the potential DNA contamination.*" in lines 574-584.

Second, we retrieved the metagenomics and metatranscriptomics data for LME and HME sheep from a published study (Shi et al. 2014) and we have provided a more detailed description of the groups. We have reorganized the Methods section to align it more closely with the Results section and have also added a more detailed description of the statistical analysis as "*Based on this study, methane measurements were conducted on 22 sheep and sorted by their mean production values. Four high emitters and four low emitters (each sampled at two time points) were selected for further analysis.*" in lines 800-802 and "*The comparison of ancestral gene families in genera from the environmental and gastrointestinal clades, as well as the assessment of taxonomic and gene abundance differences between the LME and HME groups, were calculated using the Wilcoxon rank-sum test by the wilcox.test module in R (v.4.1.3) with the parameter "paired = FALSE," and the random forest for differential gene selection was constructed with 1000 trees using the R package randomForest (v.4.7-1; <https://cran.r-project.org/web/packages/randomForest>). PCoA plot of 192 Euryarchaeota and Crenarchaeota*

genomes, as well as 243 Methanomassiliicoccales genomes, revealed the genomic relationships using the Bray-Curtis dissimilarity matrix based on the profiles of gene orthologs, and the ANOSIM test in the R package vegan [136] (v.2.5-6) with 9,999 permutations validated the differences. The correlations between Methanomassiliicoccales and other orders were calculated using the Hmisc module of R based on the Spearman correlation test with an asymptotic measure-specific P value.” in lines 822-832.

Q6. Figures - Legends needs to be bigger for all the plots, heatmaps, and schematic diagrams. Figure panels seems to be too crowded.

R: Thank you for pointing this out. We have made the necessary modifications to improve the readability of the figures and legends.

Reviewer #3

Unraveling the phylogenomic diversity of Methanomassiliicoccales: implications for mitigating ruminant methane emissions.

The widespread distribution of Methanomassiliicoccales, an order among Euryarchaeota that has the functionality to utilize “methyl” compounds (methyl amines and methanol) has created a renewed interest among scientific communities that work around anaerobic microbial communities inhabiting host/habitats over the past decade. Several papers have discussed that the “rumen environment” is no less and has a rich community of Methanomassiliicoccales (Janssen and Kirs, 2008, Poulsen et al., 2013, Borell et al., 2014, Sollinger et al., 2016, 2018 and 2019, Seedorf et al., 2015 and several others). Many of these have performed detailed comparisons of Methanomassiliicoccales distribution among the gastrointestinal tract including rumen to those of other environments. These studies highlighted the significance of adaptation of these methylotrophs to respective environments. Compared to these studies, the current study of Xie et al., is not completely novel, but an extension of the studies reported thus far. However, the authors have employed a robust analytical approach integrating multi programs and bioinformatic tools to integrate multiple metagenomic datasets from diverse environments, assemble MAGs to better understand evolutionary, phylogenetic, and functional roles of Methanomassiliicoccales with emphasis on the rumen environment. While the strength of the study is in the use of these sophisticated tools to provide new information on novel genera, genes and associations between pathways that have evolutionary significance, the study was constrained to link the findings to methane abatement. Overall, the manuscript is well written and organized into sections that are summarized well and supported with graphic representations. Because discussion is integrated within results sections, the reasoning and justification of why the authors had to perform these complicated analyses is seriously lacking. This reviewer still sees the sections as discrete but not cohesive. A major limitation I see in this study is the lack of hypotheses for each approach applied, take home message, strengths and limitations associated with each study section, and how the work is linked to methane mitigation.

R: We greatly appreciate the professional suggestions raised by the reviewer, which have helped us improve the clarity and scientific depth of this manuscript. Although this study builds upon several previous bodies of work on Methanomassiliicoccales that have predominantly utilized the 16S rRNA gene, *mcrA* gene, or cultivation-derived data to present many important findings (highlighted in the revised Introduction section), the novelty of our study lies in the extensive use of uncultured metagenome-assembled genomes (MAGs), providing new genome-centric insights into the Methanomassiliicoccales community. In recent times, hundreds of thousands of MAGs have been generated from diverse habitats, including many newly discovered species. The genomes of Methanomassiliicoccales within these datasets have not been comprehensively integrated and utilized, which we believe is crucial for advancing research on Methanomassiliicoccales. Just as you mentioned, we aimed to contribute by employing advanced analytical tools and integrating diverse metagenomic datasets to gain new insights into the evolutionary and functional aspects of Methanomassiliicoccales on a larger scale. We are pleased that this point and its significance are clear to the reviewer. In terms of methane mitigation, we believe that the Methanomassiliicoccales genera emphasized in this study are likely to serve as indicative microbes for a low-methane-emitting (LME) microbiome and may be applied in future breeding programs for ruminants. Additionally, our findings suggest that nutrient availability, such as B₁₂, plays an important role in regulating methane emissions in ruminants. In the revision, we have separated the results and discussion sections (also suggested by Reviewer #2) and

included additional descriptions and hypotheses dedicated to scientific explanations. We have also made efforts to address each of your suggestions, as detailed below.

General comments:

Q1. The first comment is that except for using metagenomic datasets, I see this study to be very similar to the efforts of Sollinger et al., 2016, and 2019. Comparisons clearly state that the diversity of Methanomassiliicoccales in the rumen or GI tracts and other environments are different. Identification of a few lineages as genera has also been discussed in Sollinger and Seedorf studies. Both 16S rDNA and McrA genes have been used to compare the novel genera in previous studies, whereas the current study used only 16S rDNA.

R: Thanks for your comment. Studies on Methanomassiliicoccales conducted by Söllinger et al. and Seedorf et al. have yielded many important findings and proposed original points, as outlined in the revised Introduction section (Seedorf et al., 2015; Söllinger et al., 2016; Söllinger et al., 2019). In this study, we aimed to harness large-scale genomic comparisons to acquire a wealth of new ecologic and metabolic information on Methanomassiliicoccales, as the tremendous potential of the metagenome to provide novel genomic insights that are not culture-dependent is becoming evident in many studies.

Regarding the methods you mentioned for identifying novel genera, we utilized whole-genome alignment-based methods to determine known or novel taxonomic units in Methanomassiliicoccales based on average amino acid identity (AAI) and average nucleotide identity (ANI). The similarity of the 16S rDNA gene was used as a reference in this study. According to your suggestion, we have also incorporated a phylogenomic analysis using the *mcrA* genes to support the taxonomy in the revised manuscript, and found that the taxonomy closely aligns with the whole-genome phylogenomic tree. We have added the results “*Maximum likelihood trees were constructed using 16S rRNA genes from 141 genomes, mcrA genes from 199 genomes and whole-genomes (n = 243), which collectively support the Methanomassiliicoccales order comprising of one family with significant variation at the genus level (Fig. 2B and Fig. S2).*” in lines 187-190 and the methods “*Likewise, the mcrA genes from 199 Methanomassiliicoccales genomes and 18 Methanosarcinales genomes (outgroup taxa) were extracted and used to construct a maximum likelihood tree based on the LG+F+I+I+R4 model.*” in lines 771-773.

Q2. Second, the functional significance of Methanomassiliicoccales in the rumen has not been discussed. While an attempt to show the distribution of methyl amines, and methanol along with cofactor metabolism in B12 synthesis have been identified to enable Methanomassiliicoccales to cluster separately, the authors did not discuss the functional relevance. Vitamin B12 is part of corrinoid proteins, the central core of methyltransferases and so discussion on methyl transferases and how these are distinct among different clades of Methanomassiliicoccales may have set a new direction for this study.

R: We appreciate the point raised by you to improve significance. We agree and have added the related results as “*B₁₂ is a major component of corrinoid proteins, forming the central core of methyltransferases (mtaC, mttC, mtbC, mtmC, and mtsB) [59]. We also observed that these B₁₂-dependent methyltransferases were more prevalent in the gastrointestinal clades (G. 18, G. 20, G. 21, and G. 22), with the exception of the human gut-derived environmental clade G. 9.*” in lines 270-273. In addition, we have included more discussions as “*Methanomassiliicoccales were found to possess genes involved in the anaerobic B₁₂ biosynthesis pathway, which is significant for their metabolic*

processes. Furthermore, the widespread presence of complete *BtuFCD* components for B_{12} transport indicated a capability to acquire and utilize this essential coenzyme. Like other nutrient metabolites studied in microbial interactions, this also implies that the transport of B_{12} from the external environment is crucial for *Methanomassiliicoccales* members. The possession of *de novo* biosynthesis and salvage routes for B_{12} in *Methanomassiliicoccales* is not surprising, as methylotrophic corrinoid proteins (*mtaC*, *mttC*, *mtbC*, *mtmC*, and *mtsB*) are homologous to the core corrinoid-binding domain of B_{12} proteins [59, 67]. This suggests that it is an important precursor for B_{12} -dependent methyltransferases by *Methanomassiliicoccales* during methanogenesis. Interestingly, the prevalence of B_{12} -dependent methyltransferases varied among different *Methanomassiliicoccales* genera, possibly reflecting their specific metabolic requirements related to B_{12} in their ecological niches.” in lines 471-482.

Q3. The third comment is while the authors have attempted to compare with *Methanobrevibacter*, why has there not been any attempts to compare this *Methanomassiliicoccales* with *Methanosphaera*? The latter methanogen belongs to *Methanobacteriales*, adapted to using methanol as the major substrate by having *mtaA*, *mtaB* and *MtaC* genes. Just curious why the study did not pick up the presence of *MtaB* and *MtaC* from *Methanosphaera*? Among methylotrophic methanogens, both *Methanosphaera* and *Methanomassiliicoccales* are considered for discussion. This paper may benefit from adding a section on comparison of *Methanosphaera* and *Methanomassiliicoccales*.

R: Thanks for this constructive suggestion. We do agree a comparison with *Methanosphaera* could provide valuable insights into the functional roles within the rumen methanogenic archaea. As part of a complete re-analysis of the Shi et al. dataset, we have added the related results as “*Notably, M. stadtmanae* belongs to the *Methanobacteriales* order, but its energy metabolism is similar to that of *Methanomassiliicoccales*, as it relies on a H_2 -dependent methylotrophic metabolism [5]. Indeed, we observed that *M. stadtmanae* possesses the *mtaB* and *mtaC* genes, which facilitate methanogenesis from methanol and H_2 (Fig. 1C).” in lines 142-146 and “*Shi et al. previously reported a higher relative abundance of metabolically similar Methanosphaera spp. in the LME sheep [53]. However, at a transcriptomic level, we observed numerically higher gene expression levels in M. stadtmanae in the LME sheep, although the differences were not statistically significant (Fig. S6B).*” in lines 354-357. We have also added the related discussions as “...Interestingly, *Shi et al. found that M. stadtmanae, which are also H₂-dependent methylotrophic methanogens, exhibited higher abundance in the LME sheep [53]; our re-analysis confirmed this observation although with no significant difference. The genera (G. 21, G. 22, and Methanosphaera spp.) with similar metabolic characteristics are likely to serve as indicative microbes for an LME microbiome and may be applied in future breeding programs for ruminants.*” in lines 520-524.

Q4. Fourth, the inclusion of metagenomic and metatranscriptomic data of high and low methane emitting sheep remains questionable. The authors have used metagenomic datasets from diverse ruminant species and described the distribution of two distinct clades: LGM-RCC1 and LGM-DZ1. None of these were identified in high or low methane emitting sheep. The genus identified was G.21 across all sheep. Except for the fact that *Methanobrevibacter* was enriched in HME compared to LME, whereas *Methanomassiliicoccales* between LME and HME sheep has not been justified very well. Shi et al 2014 described that LME may be enriched in *Methanosphaera* but there has not been an attempt to discuss connection between *Methanosphaera* and *Methanomassiliicoccales*.

R: Thank you for pointing this out. In this section, we utilized a dataset consisting of 68 genomes derived from the GITs of ruminants as our reference genome dataset and identified 12 significantly differential genomes among these 68 Methanomassiliicoccales genomes. We excluded the four MAGs recovered from the LME and HME sheep to avoid technical errors, as using MAGs directly mapped to the samples they were assembled from can make it challenging to obtain accurate abundance values. In addition, metagenomic data from the LME and HME sheep were also used to support the dominance of certain genera of Methanomassiliicoccales. We have added the related explanations and results “...*Focusing specifically on the Methanobacteria, Shi et al. revealed higher relative abundances of organisms belonging to Methanobrevibacter spp. in the HME sheep [53]. However, due to a lack of Methanomassiliicoccales genome representation at the time of the Shi et al. study, their transcriptional activity in the LME and HME animals was not recorded. Therefore, we further aligned transcriptomic reads from the LME and HME sheep to 68 ruminant-derived Methanomassiliicoccales genomes and seven publicly available genomes from the Methanobrevibacter genus....In contrast, 12 significantly differential Methanomassiliicoccales genomes ($P < 0.05$) from G. 16, G. 21, and G. 22 were mostly enriched in the LME sheep, except for RGIG2195 (G. 16) and ISO4-G1 (G. 22) with 1.5- and 2.8-fold enrichment in the HME sheep, respectively (Fig. 6B)....To advance this analysis further, we reanalyzed metagenomic data from both the LME and HME groups recovering 17 high-quality archaeal MAGs affiliated to Methanosphaera ($n = 1$), Methanomassiliicoccales ($n = 4$, classified as Methanomethylophilus in the GTDB) and Methanobrevibacter ($n = 12$) (Table S3). The phylogenomic positions of these four Methanomassiliicoccales MAGs maintained taxonomic consistency all clustering within G. 21 (Fig. 6C). In support of our hypothesis that G. 21 Methanomassiliicoccales populations are more active in the LME sheep, three of the four MAGs were obtained from the LME samples, signifying the dominance (Fig. 6C).*” in lines 343-365.

Regarding the comparison of Methanomassiliicoccales between the LME and HME sheep, we have included more discussions based on improved results as “*Methanogens play a significant role in the rumen ecosystem by efficiently consuming H_2 , thereby promoting the recycling of reduced cofactors and facilitating the breakdown and fermentation of plant materials [70, 71]. Therefore, strategies for mitigating methane emissions should consider alternative pathways for H_2 utilization to decrease methanogenesis levels while preserving rumen function [69, 72, 73]. Methanobrevibacter spp. have been confirmed as the predominant taxa responsible for high methane emissions, and they maintain a dominant presence in the rumen [53, 74]. Our re-analysis of the Shi et al. dataset [53] to incorporate Methanomassiliicoccales gene expression in sheep categorized as either high (HME) or low emitting (LME) showed that the overall Methanomassiliicoccales gene expression was higher in the HME sheep, however for several Methanomassiliicoccales genera, such as G. 21 (or Methanomethylophilus) and G. 22, their gene expression was in fact higher in the LME sheep. Interestingly, Shi et al. found that *M. stadtmanae*, which are also H_2 -dependent methylotrophic methanogens, exhibited higher abundance in the LME sheep [53]; our re-analysis confirmed this observation although with no significant difference. The genera (G. 21, G. 22, and Methanosphaera spp.) with similar metabolic characteristics are likely to serve as indicative microbes for an LME microbiome and may be applied in future breeding programs for ruminants.*” in lines 510-524.

Q5. Finally, the title of the paper is to link Methanomassiliicoccales and implications to methane mitigation. Results and discussion around methanogenesis pathways, particularly distribution of genes coding for MCR and HDR enzymes, the key enzymes in methanogenesis has been poorly described.

Shi et al 2014 highlights the distribution of *mcr*/*mrt* genes as critical for FIME and LME and that *mrt* genes (MCR isoenzyme II) may be enriched in Methanomassiliicoccales and Methanosphaera, and in general in Methylophilic methanogens. A comparison of *mcr* and *mrt* between different MAGs may add new insights on the distribution of Methanomassiliicoccales. Further, the use of Rice cluster C, the distribution of methanomassiliicoccales with and without cytochromes, and the emergence of methanomassiliicoccales without cytochromes as evolutionary younger than with cytochromes, as described in Thauer et al 2008 has not been discussed.

R: We agree with the reviewers concerns and in response have added the results on the distribution of the cytochrome-containing HdrE enzyme as “*Methylophilic methanogens can oxidize additional methyl groups to CO₂, and this capability is exclusively found among members of the Methanosarcinales [10]. As for methylophilic methanogens without cytochromes, methyl groups cannot be oxidized to CO₂; instead, they obligately use H₂ to reduce methyl compounds to CH₄ [11]. In this context, we observed a homolog of HdrE, which encodes the cytochrome b-containing membrane anchor of the heterodisulfide reductase complex (HdrDE) responsible for receiving electrons from methanophenazine, was present in all Methanosarcinales genomes and in the species Methanosphaerula palustris within Methanomicrobiales (Fig. S1). However, Methanomassiliicoccales, Methanocellales (Rice Cluster I; despite the prior discovery of cytochromes in some of the MAGs [54]) and other methanogenic orders, all lack the HdrE subunit (Fig. S1A).*” in lines 123-132 and also the discussions as “*Nonetheless, Methanomassiliicoccales, which evolved independently and lack cytochromes, emerged at a relatively later point in time compared to Methanosarcinales and do not fall into either the class I or class II methanogens. As previously discussed, methanogens possessing cytochromes are evolutionarily younger than those lacking cytochromes [5]. Furthermore, methanogens with cytochromes demonstrate notably enhanced growth yields compared to methanogens that lack cytochromes [5]. These findings may suggest that the acquisition of cytochromes in methanogens was possibly driven by environmental adaptation. This is supported by the presence of the cytochrome b subunit HdrE in the species Methanosphaerula palustris, which is the only methanogen, besides Methanosarcinales, known to encode it. Previous research has indicated that the *hdrE* gene in *M. palustris* may have been acquired through horizontal gene transfer from a Methanosaeta lineage [65].*” in lines 396-406.

Additionally, a comparison of the *mcr* and *mrt* genes assembled from the LME and HME sheep has been previously conducted by Shi et al. to determine the methanogenic lineages within the metagenomic data. The distinct clusters were subsequently used to compare their total gene abundances, and this analysis could reveal previously undefined taxa. We have incorporated phylogenomic analysis using the *mcr* gene of these Methanomassiliicoccales genomes into the revised manuscript, and found that the taxonomy closely aligns with the whole-genome phylogenomic tree in lines 187-190. This indicates that there is a high consistency on the distribution of Methanomassiliicoccales between the phylogenomic branching and the *mcr*-based branching.

Q6. The study is aimed to address an important topic that has been daunting to the ruminant systems, is to better understand the role of Methanomassiliicoccales. As discussed above, several papers have been published explaining the distribution of this relatively new clade of methanogens. While the study has invested significant efforts in assembly and comparisons, the underlying scientific description is rather weak, and will certainly benefit from revising the scientific content. A good description of hypotheses with rationale for section of analysis, followed by a separate section of discussion, highlighting the connection between the findings and methanogenesis pathways, strengths and directions for the future

are all needed to make this study more suitable for publication in Genome Biology, and to use this information for mitigation of enteric methane emissions from diverse ruminant systems.

R: Thank you for your constructive feedback. Following your suggestions, and those concerning manuscript structure by Reviewers #1 and #2, we have made careful revisions to this manuscript. We believe that the revised version has seen improvements in both scientific significance and organizational structure.

Specific comments:

Q1. Abstract: Ln 18-20: Phylogeny and ecological roles have been described in previous reports (Thauer et al 2008; Sollinger et al 2015, 2019, Seedorf et al). The authors should mention these studies, and describe what the authors wanted to accomplish through this study.

R: We agree and have described these previous studies in the revised Introduction section. This sentence has been modified to *“Methanomassiliicoccales are a recently identified order of methanogens that are diverse across global environments particularly the gastrointestinal tracts (GITs) of animals, however their metabolic capacities are defined via a limited number of cultured strains.”* in lines 17-19. Additionally, we have added the detailed objectives of this study in the revised Introduction section (lines 101-112).

Q2. Ln 21-25 has been done previously as detailed in the previous reports.

R: We agree and have revised this sentence to *“Here, we profiled and analyzed 243 Methanomassiliicoccales genomes assembled from both cultured representatives and uncultured metagenomes recovered from various anthropogenic and natural biomes, including the GITs of different animal species. Our analyses reveal the presence of numerous undefined genera and suggest genetic variability in unique metabolic capabilities within Methanomassiliicoccales lineages, which is essential for adapting to their respective ecological niches. In particular, the GIT Methanomassiliicoccales demonstrate the presence of co-diversified members with their hosts over evolutionary timescales and likely originated in the natural environment.”* in lines 20-26.

Q3. Ln 27-29: Not clear. Why wasn't the ISO4 strain isolated from sheep not used in analysis.

R: Thanks. We would like to clarify that we included both ISO4-G1 and ISO4-H5 strains in this study. Due to the limited length of the Abstract section, we decided not to provide detailed descriptions of the strains included in this study. However, you can find that ISO4-G1 and ISO4-H5 were both compared in the Results section. Additionally, reference and genomic information for all the Methanomassiliicoccales genomes used in this study were summarized in Table S3.

Q4. Ln 29-30: Vague statement.

R: We agree and have removed this sentence to include more key results in the Abstract section.

Q5. Ln 31-33: If the functional role of Methanomassiliicoccales was proposed from FIME and LME only, why was the analysis done to assemble MAGs? What is the connection between the Ln 93 to 365 and HME/LME analysis?

R: Thanks for your important question. The data of the LME and HME sheep provide a good case for validating our genomic findings in a real rumen ecosystem. We have added more descriptions to improve the connection in multiple parts of the manuscript. For example, *“Focusing specifically on the*

Methanobacteria, Shi et al. revealed higher relative abundances of organisms belonging to *Methanobrevibacter* spp. in the HME sheep [53]. However, due to a lack of *Methanomassiliicoccales* genome representation at the time of the Shi et al. study, their transcriptional activity in the LME and HME animals was not recorded. Therefore, we further aligned transcriptomic reads from the LME and HME sheep to 68 ruminant-derived *Methanomassiliicoccales* genomes and seven publicly available genomes from the *Methanobrevibacter* genus.” in lines 343-349, “In support of our hypothesis that *G. 21 Methanomassiliicoccales* populations are more active in the LME sheep, three of the four MAGs were obtained from the LME samples, signifying the dominance (Fig. 6C).” in lines 363-365, and “In light of this, we examined the B_{12} biosynthesis pathway in the rumen microbiome and observed elevated levels of gene expression in multiple processes (*gltB*, *gltD*, *glnA*, *MET8*, *hemA*, and *cbiL*) within the LME sheep (Fig. S6D). Furthermore, we also observed a significant increase in the total gene expression of the vital B_{12} transporter *BtuC* (*btuC*, $P < 0.001$; Fig. 6D) in the rumen microbiome of the LME sheep.” in lines 374-378.

Q6. Ln 34-36: Conclusion is not well justified.

R: We agree and have revised this sentence to “We provide a comprehensive and updated account of divergent *Methanomassiliicoccales* lineages, drawing from numerous uncultured genomes obtained from various habitats. We also highlight their unique metabolic capabilities involving B_{12} , which could serve as promising targets for mitigating ruminant methane emissions by altering H_2 flow.” in lines 33-36.

Q7. Introduction: This section requires major revision. The description of *Methanomassiliicoccales* in diverse environments and the rumen is very shallow. Several papers have been published attempting to understand the role of this clade. Discussion of those papers is needed.

R: We agree with both this comment and those made by Reviewer #1 and have revised the Introduction section to emphasize the significance of *Methanomassiliicoccales* and the progress in previous research (please see below).

“The *Methanomassiliicoccales* order was previously described as ‘rice cluster C’ or ‘rumen cluster C’ and is distantly related to *Thermoplasmatales* [15-17]. They have been discovered in various anoxic environments, including wetlands, oceans, deep subsurface, rice crops, and the gastrointestinal tracts (GITs) of both humans and several animals [12, 15, 18, 19]. Paul et al. originally characterized this group by observing an animal-associated cluster distinct from the environmental cluster through the analysis of comparative 16S rRNA and methyl coenzyme M reductase alpha subunit (*mcrA*) genes [15]. Söllinger et al. have subsequently surveyed the physiology and ecology of *Methanomassiliicoccales*, highlighting their widespread distribution in global wetlands and animal GITs, as well as their significant contribution to methane emissions from these climate-relevant ecosystems [12]. The first *Methanomassiliicoccales* isolate, *Methanomassiliicoccus luminyensis*, was reported by Dridi et al. to have been obtained from human feces [20]. In the past decade, despite many efforts, only eight complete genomes of *Methanomassiliicoccales* isolated from the GITs of various animals have been characterized through pure cultures, including those from humans [21, 22], termites [23], chickens, sheep [24, 25], and cattle. The cultivated representatives of *Methanomassiliicoccales* have demonstrated an energy metabolism distinct from that of other methylotrophic methanogens [5, 23, 26], and have been found to utilize methylamines and methanol as electron acceptors [13, 15, 20, 21, 27, 28]. Genomic analyses

conducted thus far have revealed that *Methanomassiliicoccales* lack cytochromes and encode a truncated Wood-Ljungdahl (WL) pathway [11, 23, 26, 29]. In this context, methyl groups cannot be oxidized to CO₂; instead, they obligately employ H₂ to reduce methyl compounds to CH₄ [10, 11]. Like *Methanimicrococcus blatticola* and *Methanosphaera* species [5], *Methanomassiliicoccales* rely on external H₂ and have become a model for H₂-dependent methylotrophic methanogens due to their extensive distribution [12].” in lines 52-73.

“The occurrence of *Methanomassiliicoccales* in the human gut has been linked to lowered trimethylamine oxide (TMAO) production and cardiovascular disease risk [34, 44]. As for other animals, such as ruminants, where the metabolism of methanogens contribute around 37% of the world’s total anthropogenic methane emissions [27, 45], the *Methanomassiliicoccales* order has been relatively poorly characterized [17]. A global rumen survey conducted by Henderson et al. previously suggested that ruminal methanogens is limited to the phylum *Euryarchaeota* [46]. However, over the last decade, *Thermoplasmatales*-affiliated sequences (i.e., *Methanomassiliicoccales*) have been found in a variety of ruminant species and often constitute the second most abundant group of methanogens found in the rumen microbiome [17, 27, 47-52]. More recently, Shi et al. conducted a comparative analysis of the ruminal archaeal microbiota in sheep with low and high methane emissions, although *Methanomassiliicoccales* was not included due to a lack of genomic information at the time of analysis [53]. Instead, it was shown that *Methanosphaera* spp., which shares an energy metabolism similar to *Methanomassiliicoccales*, exhibited elevated abundance in sheep with low methane emissions [53]. Overall, the role of *Methanomassiliicoccales* in the rumen remains unclear due to a lack of sufficient phylogenetic classification, and further analysis specifically focused on *Methanomassiliicoccales* is needed in light of this previous research deficiency.” in lines 85-100.

Q8. Ln 67-68: The role of *Methanomassiliicoccales* in methane emissions is still speculative.

R: We agree and have revised this sentence as “As for other animals, such as ruminants, where the metabolism of methanogens contribute around 37% of the world’s total anthropogenic methane emissions [27, 45], the *Methanomassiliicoccales* order has been relatively poorly characterized [17].” in lines 86-89.

Q9. Ln 70-74: Link to TMAO is left hanging and is not connected with the rest of the section.

R: Thanks for point out this. We have revised this sentence as “The occurrence of *Methanomassiliicoccales* in the human gut has been linked to lowered trimethylamine oxide (TMAO) production and cardiovascular disease risk [34, 44].” and relocated it to lines 85-86.

Q10. Hypotheses and objective(s) must be stated clearly and how the authors planned to accomplish.

R: We agree and have added the specific hypotheses and objectives of this study as “Here, we collected 208 publicly available *Methanomassiliicoccales* genomes and MAGs from various global habitats as well as reconstructed 33 high-quality *Methanomassiliicoccales*-affiliated MAGs. MAGs were recovered from 370 metagenomic datasets that were generated across 10 GIT regions of seven different ruminant species and eight metagenomic samples enriched with trimethylamine from cow rumen fluids. On our newly acquired genome atlas, we applied a large-scale genome-resolved comparison to reveal the essential metabolic functions of *Methanomassiliicoccales* and clarify the evolutionary classification of *Methanomassiliicoccales* originating from the GITs of ruminants. Subsequently, a genome-centric

isolation strategy and in vitro enrichments of samples collected from the goat rumen and cow abomasum obtained pure culture strains of the major Methanomassiliicoccales groups. Finally, we additionally integrated metagenomic and metatranscriptomic data from Shi et al. [53] to provide novel insights into the distinct Methanomassiliicoccales groups in the rumen and their roles in methanogenesis.” in lines 101-112.

Q11. Results and Discussion

Ln 92-109: Why was this analysis done?

R: Thanks for this question. This analysis clarifies the phylogenetic position of Methanomassiliicoccales among these archaeal orders within Euryarchaeota and serves as the basis for our subsequent investigation into the evolutionary analysis and gastrointestinal adaptations. To enhance clarity regarding our objectives, we have made a revision and included an explanation “*To determine the phylogenetic position of the relatively new order Methanomassiliicoccales, we investigated the evolutionary divergence of the Euryarchaeota using publicly available datasets in National Center for Biotechnology Information (NCBI) database, including 146 representative genomes from 13 orders of Euryarchaeota and 46 outgroup genomes from Crenarchaeota (Table S1).*” in lines 115-118.

Q12. Ln 99-102: sentence must be revised.

R: We agree and have revised this sentence to “*The genome-wide phylogenetic tree constructed using completed genomes from the Euryarchaeota phylum revealed that Thermococcales represented the deepest root among these orders, suggesting that methanogenesis may not be an ancestral metabolic process in Euryarchaeota [63]. This implies that the methanogenesis capabilities of these methanogens have evolved gradually throughout their evolutionary history. Indeed, methanogens within Euryarchaeota can be divided into two clades: class I (Methanococcales, Methanobacteriales, and Methanopyrales) and class II (Methanosarcinales, Methanocellales, and Methanomicrobiales).*” in lines 388-394.

Q13. Ln:107-109: with and without cytochromes may explain their evolution as described in Thauer et al 2008.

R: Thanks for your suggestion. Based on the description by Thauer et al., we have introduced a discussion regarding the potential contribution of cytochromes to these evolutionary differences as “*Nonetheless, Methanomassiliicoccales, which evolved independently and lack cytochromes, emerged at a relatively later point in time compared to Methanosarcinales and do not fall into either the class I or class II methanogens. As previously discussed, methanogens possessing cytochromes are evolutionarily younger than those lacking cytochromes [5]. Furthermore, methanogens with cytochromes demonstrate notably enhanced growth yields compared to methanogens that lack cytochromes [5]. These findings may suggest that the acquisition of cytochromes in methanogens was possibly driven by environmental adaptation. This is supported by the presence of the cytochrome b subunit HdrE in the species Methanosphaerula palustris, which is the only methanogen, besides Methanosarcinales, known to encode it. Previous research has indicated that the hdrE gene in M. palustris may have been acquired through horizontal gene transfer from a Methanosaeta lineage [65].*” in lines 396-406.

Q14. Ln 110-148: what about Methanospaera? Although Methanobacteriales, it does have the mtaA,

mtaB and mtaC and vit B12. Why was this not described? Also what is the involvement of cytochromes if it is related to evolution?

R: Thanks for your suggestion. We have now added the results regarding *Methanosphaera* as “Notably, *M. stadtmanae* belongs to the Methanobacteriales order; but its energy metabolism is similar to that of Methanomassiliicoccales, as it relies on a H₂-dependent methylotrophic metabolism [5]. Indeed, we observed that *M. stadtmanae* possesses the mtaB and mtaC genes, which facilitate methanogenesis from methanol and H₂ (Fig. 1C).” in lines 142-146. Additionally, we have conducted an analysis and included the results related to cytochrome-containing HdrE enzyme as “Methylotrophs with cytochromes can oxidize additional methyl groups to CO₂, and this capability is exclusively found among members of the Methanosarcinales [10]. As for methylotrophs without cytochromes, methyl groups cannot be oxidized to CO₂; instead, they obligately use H₂ to reduce methyl compounds to CH₄ [11]. In this context, we observed a homolog of HdrE, which encodes the cytochrome b-containing membrane anchor of the heterodisulfide reductase complex (HdrDE) responsible for receiving electrons from methanophenazine, was present in all Methanosarcinales genomes and in the species *Methanosphaerula palustris* within Methanomicrobiales (Fig. S1). However, Methanomassiliicoccales, Methanocellales (Rice Cluster I; despite the prior discovery of cytochromes in some of the MAGs [54]) and other methanogenic orders, all lack the HdrE subunit (Fig. S1A).” in lines 123-132. We have also added a corresponding discussion section in lines 396-406.

Q15. Ln 170-196: This section is interesting and provides insights on distribution across diverse environments. What genera were abundant among ruminant datasets?

R: Thank you for your positive comments. The results showed that G. 21 and G. 22 possess the most abundant assembled MAGs originating from the GITs of ruminants, and they represent the dominant genera among ruminant datasets. We have added this results in the revision “A phylogenetic tree based on a total of 68 Methanomassiliicoccales genomes from ruminant GITs showed that these 33 MAGs formed six distinct genera, including G. 15, G. 16, G. 17, G. 20, G. 21, and G. 22, and were assigned to 23 species (Fig. 5B and Table S3). The most abundant genera were G. 21 and G. 22, indicating their significance in methane production in ruminants.” in lines 285-288.

Q16. Ln 197-247: This is a good exercise but the use of this information for cellular metabolism and methane emission is needed.

R: We agree with your perspective and have added some connections with the next metabolic section into the Discussion section “The findings from functional analysis suggested that the evolutionary divergence between genera may signify the emergence of novel genes. ... Overall, genetic variability has led to functional partitioning and ecological divergence among Methanomassiliicoccales lineages, resulting in distinctive metabolic capabilities crucial for their respective ecological niches.” in lines 445-456.

Q17. Ln 248-292: Variations based on methyltransferases, mcr/mrt and HdrABC and HdrDE may provide better insights to link the information to methanogenesis.

R: Thanks for your suggestion. We have revised this section to present more association with methanogenesis “Core methanogenesis genes (mcrABCDG, mtbA, and mtaA) were also found in Methanomassiliicoccales, whereas the mtmC gene, which encodes an enzyme that catalyzes the transfer of a methyl group from monomethylamine to monomethylamine-specific corrinoid proteins, was missing

from all genomes (**Fig. 4**). We found that all the *Methanomassiliicoccales* genomes lack the homologs of the *HdrE* subunit (*hdrE*) but carry the *Fpo*-like subunits (*fpoKLMN*), which may directly interact with the *HdrD* subunit (*hdrD*) to form the energy-converting ferredoxin:heterodisulfide oxidoreductase complex (**Fig. 4**). Moreover, *Methanomassiliicoccales* possessed a large amount of methyl viologen-dependent hydrogenase (*mvhADG*) and the *Hdr* complex (*hdrABC*) to couple the reduction of the heterodisulfide CoM-S-S-CoB and a ferredoxin with H_2 via flavin-based electron bifurcation (**Fig. 4**). ... B_{12} is a major component of corrinoid proteins, forming the central core of methyltransferases (*mtaC*, *mttC*, *mtbC*, *mtmC*, and *mtsB*) [59]. We also observed that these B_{12} -dependent methyltransferases were more prevalent in the gastrointestinal clades (G. 18, G. 20, G. 21, and G. 22), with the exception of the human gut-derived environmental clade G. 9.” in lines 254-273.

Q18. Ln LGM-DZI survives in acidic environment (abomasum). Are these detected in the ruminal environment?

R: Thanks for this question. In our evolutionary analysis, we observed that LGM-DZ1 clusters with the majority of MAGs originating from the rumen. Moreover, LGM-DZ1 and related uncultured rumen MAGs exhibited genes associated with acid stress resistance. This suggests that they represent a cluster existing in both the rumen and abomasum. In the revised manuscript, we have added a discussion as “...Additionally, LGM-DZI was the first strain isolated from the abomasum and established that *Methanomassiliicoccales* could survive in the acidic environment of the digestive tract. It formed an independent branch with 13 MAGs, although interestingly the majority of them were derived from the rumen....” in lines 502-505.

Q19. Ln 366-396: This section on HME/LME appears disconnected with the remaining text.

R: We agree and have added a discussion to improve the connection as “Our aforementioned genomic comparisons conducted in *Methanomassiliicoccales* revealed that B_{12} biosynthesis and transport pathways are prevalent across various genera. Since B_{12} serves as a crucial coenzyme mainly synthesized by bacteria [59], we speculate that this higher genomic capability to metabolize B_{12} may confer a competitive advantage to the *Methanomassiliicoccales* in the LME sheep. The increased expression of cobalt transporters in the LME sheep supports that a substantial number of components required for B_{12} biosynthesis can be efficiently transported into microbial cells in the rumen when methane emissions are low. In correspondence, we also observed that multiple genes involved in the B_{12} biosynthesis pathway in the rumen microbiome were upregulated in the LME sheep. Furthermore, the elevated gene expression of the *BtuC* transport protein in the LME sheep suggests that *Methanomassiliicoccales*-affiliated organisms with a higher demand for B_{12} are more active under low-methane production conditions. B_{12} interactions among rumen microbes could serve as a crucial mechanism, enabling specific members of *Methanomassiliicoccales* to inhabit ecological niches and thereby alter the direction of H_2 . This points towards nutrient availability (such as B_{12}) playing an important role in regulating methane emissions in ruminants [59, 75-77].” in lines 525-538.

Q20. Ln 404-406: The statement is not well supported with any data.

R: For accuracy and cautious, we have rephrased the statement “Additionally, an integrated genome-centric metatranscriptomic analysis revealed that several genera of *Methanomassiliicoccales* were more transcriptionally active in low-emitting animals, where elevated gene expressions of B_{12} synthesis and transport may be associated with greater B_{12} sharing in the LME sheep.” in lines 548-551.

Q21. Ln 408: The role of Methanobrevibacter in high methane emissions has already been reported in Shi et al and does not connect with this study.

R: We have removed this sentence, adding key findings regarding Methanomassiliicoccales between the LME and HME sheep in lines 548-551.

Q22. Ln 415-475: The authors have done multiple steps in collecting natural digesta samples from different ruminant species, performed enrichment and followed by metagenomic analysis. This entire work was not emphasized nowhere in the introduction or results/discussion. This reviewer was confused between what was downloaded from public databases vs. what has been generated in-house. Coming back to my general comment, the authors could have generated multiple hypotheses from the type of work done. Culture vs. metagenomics done in house vs. publicly downloaded datasets vs. HME and LME sheep.

R: We acknowledge there exists multiple layers to our analyses however argue that all contribute towards an improved local and global perspective on Methanomassiliicoccales structure and function. To alleviate the reviewers concerns we have attempted to incorporate a clearer description into the Introduction section “*Here, we collected 208 publicly available Methanomassiliicoccales genomes and MAGs from various global habitats as well as reconstructed 33 high-quality Methanomassiliicoccales-affiliated MAGs. MAGs were recovered from 370 metagenomic datasets that were generated across 10 GIT regions of seven different ruminant species and eight metagenomic samples enriched with trimethylamine from cow rumen fluids.*” in lines 101-105 and the Discussion section “*...In this study, we collected MAGs from various habitats, reassembled high-quality MAGs from the GITs of diverse ruminant animals, cultivated new methanogenic strains of Methanomassiliicoccales, and conducted an integrated comparative genomic analysis to gain novel insights into this order.*” in lines 384-387. Additionally, in line with your suggestions, we have included some important comparisons, findings, and discussions, and attempted to interconnect them.

Q23. Better description of TMA enrichment and rationale for enrichment is needed. Why only TMA, and why not DMA and MMA and also methanol?

R: Thanks for pointing out this confusion. In the preliminary experiments, we observed that the enrichment effect of TMA appeared to be better than other methyl compounds. We have added a description of the reasons for this “*We observed in preliminary experiments that trimethylamine (TMA) served as a more effective substrate for the enrichment of Methanomassiliicoccales. Therefore, Methanomassiliicoccales were enriched with TMA from cow rumen fluids using a modified BRN medium as described in the subsequent culture section [78].*” in lines 556-559.

Q24. Ln 574-580: what is the need for rumen and abomasum contents? Methane is generated only from the rumen? Were there any new isolates of methanogens?

R: Thanks for your questions. In each experiment, we require approximately 5 mL of rumen fluids or 0.3 g of abomasum contents, which are then diluted with 50 mL of anaerobic diluting solution (lines 609-611). Methane production in ruminants has been reported to occur in the fermentative compartments (forestomach and large intestine) of their GITs, with the rumen serving as the primary site. In this study, we obtained 33 high-quality Methanomassiliicoccales MAGs from multiple GIT regions by analyzing 370 metagenomic samples from 10 gastrointestinal regions in seven ruminant

species. In addition, we were successful in isolating and cultivating two new strains, one from the goat's rumen (LGM-RCC1) and the other from the cow's abomasum (LGM-DZ1).

Q25. Ln 655: It is often suggested that while TCD can detect methane, FID provides accurate measurements on methane emissions. A suggestion to use FID specifically using on methanogenic isolates.

R: We agree your comment. While it's true that TCD may not be as accurate as FID, it is still widely used. In this study, we found that TCD was sufficient for methane detection, and the levels of methane detected were quite high.

Q26. Ln 664: typo

R: Corrected as suggested.

Q27. Figure 6: What is the unit TME stand for in the plot for bacterial and archaeal gene expression.

R: Thanks for this question. In this study, we utilized TPM (Transcripts Per Million) as the unit of gene expression, with corrections for variations in gene length and mapped reads per sample. To make it clearer, we have added a description in the Methods section "*The TPM value for a gene represents the estimated number of transcripts per million reads in the sample, considering gene length and total sequencing depth [134].*" in lines 817-818.

Q28. Figure 6C: In Shi et al. 4 HME and 4 LME sheep were selected. In the current study, 8 animals each are selected. Is it possible to add methane emissions to each of these sheep as there appears to be a large variation in the distribution of Methanobrevibacter and RGM groups. Again it is surprising that Methanosphaera is not listed as Shi et al indicated that LME sheep have been enriched in Methanosphaera.

R: Thanks for your comments. In the study by Shi et al., 4 HME sheep and 4 LME sheep were used, but each sheep was sampled at two different time points. Therefore, Shi et al. have selected 8 samples for each group as described "*Based on this study, methane measurements were conducted on 22 sheep and sorted by their mean production values. Four high emitters and four low emitters (each sampled at two time points) were selected for further analysis.*" in lines 800-802. We agree the variation between samples and have incorporated methane emissions data for each sample in the revised manuscript (**Fig. 6A**).

In the revision, we have also included the results on *Methanosphaera* as "*Shi et al. previously reported a higher relative abundance of metabolically similar Methanosphaera spp. in the LME sheep [53]. However, at a transcriptomic level, we observed numerically higher gene expression levels in M. stadtmanae in the LME sheep, although the differences were not statistically significant (Fig. S6B).*" in lines 354-357 and the discussion as "*...Interestingly, Shi et al. found that M. stadtmanae, which are also H₂-dependent methylotrophic methanogens, exhibited higher abundance in the LME sheep [53]; our re-analysis confirmed this observation although with no significant difference. The genera (G. 21, G. 22, and Methanosphaera spp.) with similar metabolic characteristics are likely to serve as indicative microbes for an LME microbiome and may be applied in future breeding programs for ruminants.*" in lines 520-524.

References

1. Marin, J., Battistuzzi, F. U., Brown, A. C. & Hedges, S. B. The timetree of prokaryotes: new insights into their evolution and speciation. *Mol. Biol. Evol.* **34**, 437-446 (2017).
2. Padmanabha, J., Liu, J., Kurekci, C., Denman, S. E. & McSweeney, C. S. A methylotrophic methanogen isolate from the Thermoplasmatales affiliated RCC clade may provide insight into the role of this group in the rumen. *Proceedings of the 5th Greenhouse Gases and Animal Agriculture Conference* p. 259 (2013).
3. Seedorf, H., Kittelmann, S. & Janssen, P. H. Few highly abundant operational taxonomic units dominate within rumen methanogenic archaeal species in New Zealand sheep and cattle. *Appl. Environ. Microbiol.* **81**, 986-995 (2015).
4. Shi, W. et al. Methane yield phenotypes linked to differential gene expression in the sheep rumen microbiome. *Genome Res.* **24**, 1517-1525 (2014).
5. Sokolovskaya, O. M., Shelton, A. N. & Taga, M. E. Sharing vitamins: cobamides unveil microbial interactions. *Science* **369**, eaba0165 (2020).
6. Söllinger, A. et al. Phylogenetic and genomic analysis of Methanomassiliicoccales in wetlands and animal intestinal tracts reveals clade-specific habitat preferences. *FEMS Microbiol. Ecol.* **92**, fiv149 (2016).
7. Söllinger, A. & Urich, T. Methylotrophic methanogens everywhere—physiology and ecology of novel players in global methane cycling. *Biochem. Soc. Trans.* **47**, 1895-1907 (2019).
8. Thauer, R. K., Kaster, A. K., Seedorf, H., Buckel, W. & Hedderich, R. Methanogenic archaea: ecologically relevant differences in energy conservation. *Nat. Rev. Microbiol.* **6**, 579-591 (2008).
9. Ungerfeld, E. M., Rust, S. R., Boone, D. R. & Liu, Y. Effects of several inhibitors on pure cultures of ruminal methanogens. *J. Appl. Microbiol.* **97**, 520-526 (2004).

Second round of review

Reviewer 1

Authors have addressed my concerns and manuscript has great improved. I have no more comments.

Reviewer 2

The authors have greatly improved the clarity of this manuscript. I could see a clear rationale with a well-justified background to the current study. Results section is much easier to follow this time. The clarity of discussion has increased greatly. I would suggest authors to be even more concise with their discussion by only focusing on the most important findings of the study. As a reader, I really want to focus on the current study outcomes and how they can be beneficial in future research. especially, where should future archaea research needs to be focused on. In addition, I recommend a professional proof reading for the language.

Authors Response

Point-by-point responses to the reviewers' comments:

Reviewer #1 Authors have addressed my concerns and manuscript has great improved. I have no more comments.

R: Thanks to the reviewer for your valuable suggestions. We appreciate your time and effort.

Reviewer #2 The authors have greatly improved the clarity of this manuscript. I could see a clear rationale with a well-justified background to the current study. Results section is much easier to follow this time. The clarity of discussion has increased greatly. I would suggest authors to be even more concise with their discussion by only focusing on the most important findings of the study. As a reader, I really want to focus on the current study outcomes and how they can be beneficial in future research. especially, where should future archaea research needs to be focused on. In addition, I recommend a professional proof reading for the language.

R: We appreciate your time in reviewing our manuscript and offering these constructive comments. In response, we have condensed the discussion section to underscore key results in the revised manuscript. In this study, we utilized numerous uncultured Methanomassiliicoccales genomes to provide novel insights into the classification, evolution, and metabolic functions of this newly identified order. Among these findings, we regard our discovery as an improved understanding of the potential impact of nutrient availability, such as B12, on the regulation of methane emissions in ruminants. Furthermore, we engaged a professional language company to proofread the entire manuscript, as per your suggestion.